# MAPK inhibitor sensitivity scores predict sensitivity driven by the immune infiltration in pediatric low-grade gliomas

Pediatric low-grade gliomas (pLGG) show heterogeneous responses to MAPK inhibitors (MAPKi) in clinical trials. Thus, more complex stratification biomarkers are needed to identify patients likely to benefit from MAPKi therapy. Here, we identify MAPK-related genes enriched in MAPKi-sensitive cell lines using the GDSC dataset and apply them to calculate class-specific MAPKi sensitivity scores (MSSs) via single-sample gene set enrichment analysis. The MSSs discriminate MAPKi-sensitive and non-sensitive cells in the GDSC dataset and significantly correlate with response to MAPKi in an independent PDX dataset. The MSSs discern gliomas with varying MAPK alterations and are higher in pLGG compared to other pediatric CNS tumors. Heterogenous MSSs within pLGGs with the same MAPK alteration identify proportions of potentially sensitive patients. The MEKi MSS predicts treatment response in a small set of pLGG patients treated with trametinib. High MSSs correlate with a higher immune cell infiltration, with high expression in the microglia compartment in single-cell RNA sequencing data, while low MSSs correlate with low immune infiltration and increased neuronal score. The MSSs represent predictive tools for the stratification of pLGG patients and should be prospectively validated in clinical trials. Our data supports a role for microglia in the response to MAPKi.

Pediatric low-grade gliomas (pLGG) are the most common brain tumors in children[1]. They comprise a variety of entities[2] and are defined as grade 1 or 2 by the World Health Organization (WHO)[3]. They are characterized by a generally favorable outcome, with a 20-year overall survival (OS) ranging from 80 to 90%[4]. Event-free survival, however, is low when incompletely resected[5], and 10-year progression-free survival ranges between 40 and 60% after adjuvant therapy[6]. This makes pLGG an often chronic progressive disease, with some patients needing several lines of systemic therapy and accumulating therapy-related sequelae[5].

In the past decade, the mitogen-activated protein kinase (MAPK) pathway was identified as the main driving force in pLGG[7], such that they are now considered to be a single-pathway-driven disease, with virtually all driving alterations occurring mutually exclusively in the MAPK pathway[8]. The most common alterations are *KIAA1549:BRAF* fusions (~50%), *BRAF*[V600E] mutations (~10%), NF1 mutations (~15%) and FGFR1/2 mutations (~10%)[2].

Since its first discovery more than 30 years ago[9], the MAPK pathway has been thoroughly described, and many inhibitors of its main mediators (i.e., BRAF[10], MEK[11], ERK[12]) have been synthesized and characterized[13]. Many inhibitors have been approved by the FDA to treat MAPK-driven diseases, such as the MEK inhibitor (MEKi) trametinib[14], binimetinib[15], selumetinib[16] and cobimetinib[17], and the type I1/2 BRAF inhibitor (BRAFi) encorafenib[18], dabrafenib[19] and vemurafenib[20]. Consequently, several (pre-)clinical studies were initiated to test the efficiency of such MAPK inhibitors (MAPKi) in pLGG. Studies using the MEKi selumetinib[21] and trametinib[22] and the BRAFi dabrafenib[23,24] showed very encouraging results, although with less extensive or durable responses than initially hoped for. Overall, 33–50% of treated patients had at least a partial response (PR) of their tumor to the treatment. Stable disease (SD) was observed in a further substantial fraction (24–67%), while progressive disease (PD), despite treatment, was observed in 4–28% of patients[21–24]. This highlights the

✉ e-mail: r.sigaud@kitz-heidelberg.de; t.milde@kitz-heidelberg.de

fact that patients harboring a pLGG will not respond homogeneously to a MAPKi despite sharing a common MAPK alteration. This is consistent with studies conducted in other MAPK-driven entities, mainly KRAS-driven tumors, where it was shown that MAPK alteration status alone does not necessarily correlate with MAPKi response[25–27]. For instance, it was shown that the treatment of MAPK pathway-driven tumors with MEKi can induce the activation of parallel pathways (e.g., receptor tyrosine kinase[28], AKT pathway[29]) via the release of negative feedback regulation of upstream partners, such as CRAF[30]. There is therefore a need to improve the stratification of patients based on their potential sensitivity to a given MAPKi in order to treat them with the therapy with the highest probability of response, i.e., significant tumor volume reduction.

Cell lines harboring the same MAPK alteration possess different levels of MAPK pathway activation[31]. This highlights the potential role that overall MAPK pathway activity level may have in predicting MAPKi sensitivity in pLGG, as opposed to genetic MAPK alteration status, which has been shown to be a poor predictor of MAPKi sensitivity[32]. In addition, phospho-ERK (pERK) is often used as a surrogate marker to assess MAPK pathway activity on formalin-fixed paraffin-embedded (FFPE) samples and to predict MAPK pathway activity levels in the clinical setting[33]. However, an increasing number of studies showed that pERK is not an accurate readout to asses MAPK pathway activity[31,34], most likely because of negative feedback loops triggered by ERK activity and involving the DUSP and SPRTY family proteins[35]. Instead, investigators suggested that measuring MAPK pathway output by expression of key MAPK target genes, such as the MAPK Pathway Activity Score (MPAS), derived from aggregated gene expression of 10 genes that have been reported in multiple gene signatures predictive of sensitivity to MAPK inhibition, would be a more accurate way to assess MAPK pathway activity[35,36].

As gene expression data from clinical sequencing analyses of patients' tumors are becoming a routine diagnostic method in precision oncology, gene expression-derived signatures are becoming more readily available as a new data type suitable for clinical applications such as predictive stratification. The pediatric precision oncology registry INFORM has shown the potential of using Next-Generation Sequencing (NGS) approaches to provide a more precise diagnosis by identifying driving alterations, thus refining treatment and improving progression-free survival[37]. These NGS techniques are being used in current recruiting and upcoming clinical trials, such as the LOGGIC/FIREFLY-2 trial[38] and its associated LOGGIC Core BioClinical Data Bank[39], generating valuable tumor-derived sequencing data. Using transcriptome signatures that could predict sensitivity to several types of drugs is therefore a realistic and clinically relevant stratification approach that could be used prospectively in upcoming clinical trials. Several studies have shown the relevance of gene expression-based signatures from cell lines in the accurate prediction of drug sensitivity[31,40,41]. Initiatives such as the Genomics of Drug Sensitivity in Cancer (GDSC) study, in which sensitivity to over a hundred compounds was measured in over 900 cell lines with known gene expression profiles, greatly helped develop such drug sensitivity signatures[42].

In this work, we show that drug class-specific MAPKi sensitivity scores (MSS) derived from gene expression signatures predict MAPKi sensitivity in pLGG in vitro and in vivo, warranting further validation in clinical trials. Our data also supports a role for microglia in the response to MAPKi.

## Results

### MAPK pathway activity, as measured by MPAS, is insufficient to predict sensitivity to all MAPKi classes in MAPK-altered cell lines
We first investigated the differential MAPK activation level in pLGG using the MPAS measure in the Open Pediatric Brain Tumor Atlas (OPBTA) dataset (Supplementary Fig. S1a). The MPAS showed a large

variability within each of the pLGG molecular subgroups (Supplementary Fig. S1b), indicating that MAPK pathway activity can differ within pLGGs harboring identical MAPK driving alterations. While all pLGG samples with a MAPK alteration had an MPAS significantly higher than the pLGG samples with a wild-type MAPK pathway and normal tissue, the $BRAF^{V600E}$ pLGG samples had the highest MPAS levels.

In order to determine how MAPK pathway activity levels correlate with MAPKi sensitivity, we correlated the MPAS with MAPKi sensitivity data (i.e., $IC_{50}$ z-scores) from MAPK-altered cancer cell lines ($n = 234$) from the Genomics of Drugs Sensitivity in Cancer (GDSC) dataset. This showed a moderate positive correlation ($r > 0.3$) of the MPAS with $IC_{50}$ z-scores for the MEKi and the BRAFi type I 1/2 and weak positive correlation coefficients ($r < 0.3$) with the BRAFi type II and ERKi (Supplementary Fig. S2). Taken together, these data indicate that MAPK pathway activity as measured by the MPAS might be suboptimal to predict sensitivity to MAPKi, in particular to BRAFi and ERKi. We hypothesized that class-specific signatures may be preferable to increase the prediction's efficiency.

### Generation of drug class-specific gene signatures to calculate MAPKi sensitivity scores (MSSs)
Class-specific MAPKi sensitivity gene signatures were generated from the GDSC dataset[42], which contains gene expression data of cell lines paired with known response to several MAPKi in vitro. To stay true to the pLGG biology, only cell lines with mutually exclusive MAPK alterations, and for which MAPKi $IC_{50}$ z-scores were available, were selected (Supplementary Fig. S3a). The dataset was then split into a Discovery set and a Testing set, ensuring equal proportions in terms of tumor of origin (Fig. 1a), samples sensitive (Fig. 1b) and resistant (Fig. 1c) to a given MAPKi, number of samples treated with the same MAPKi (Fig. 1d) and MAPK alterations (Fig. 1e, f). The Discovery set was used to generate the signatures, and the Testing set to validate them (Fig. 1g). We excluded data from cell lines treated with a MAPKi (1) without a clearly characterized mode of action to avoid drug misclassification, (2) that have already been excluded from clinical trials, (3) with target inhibition reached at concentrations higher than 100 nM, i.e., with an increased risk of off-target effects, and (4) for which a more advanced derivative was already included in the dataset (Supplementary Data 1).

For each MAPKi ($n = 9$; BRAFi Type I 1/2: dabrafenib, PLX-4720; BRAFi Type II: AZ628; MEK1/2i: PD0325901, selumetinib, refametinib, trametinib; catalytic ERKi (catERKi): ulixertinib; dual inhibitor ERKi (dualERKi): SCH-772984), samples from the Discovery set were grouped together based on their drug sensitivity, as determined by their $IC_{50}$ z-score across the entire GDSC dataset (Supplementary Fig. S3b): samples with an $IC_{50}$ z-score < −2 were considered sensitive, while cell lines with an $IC_{50}$ z-score >0 were considered non-sensitive (see Supplementary Fig. S4a for more details). Sensitive and non-sensitive groups were subjected to a Gene Set Enrichment Analysis (GSEA). The genes contributing to the enrichment edge of the "Hallmark_KRAS_SIGNALING_UP" signature in the sensitive subset were used as signatures. This was done to avoid the introduction of biases from background genes related to the tissue of origin the cell lines were derived from (e.g., melatonin production-related genes from tumor cells derived from melanoma) and only keep MAPK-relevant genes. This generated one signature per MAPKi, i.e., nine signatures related to MAPKi sensitivity (Supplementary Data 2).

These signatures were used in a single-sample gene set enrichment analysis (ssGSEA) to calculate corresponding predictive sensitivity scores in the Testing set. The prediction ability of those scores was evaluated in the Testing set (Supplementary Fig. S3c), using five performance metrics: (1) Youden's J stat and (2) F1-score, from a Receiver operating characteristic (ROC) analysis for the scores to predict MAPKi sensitivity; (3) Pearson's correlation coefficient, measuring the correlation between the scores and the measured MAPKi

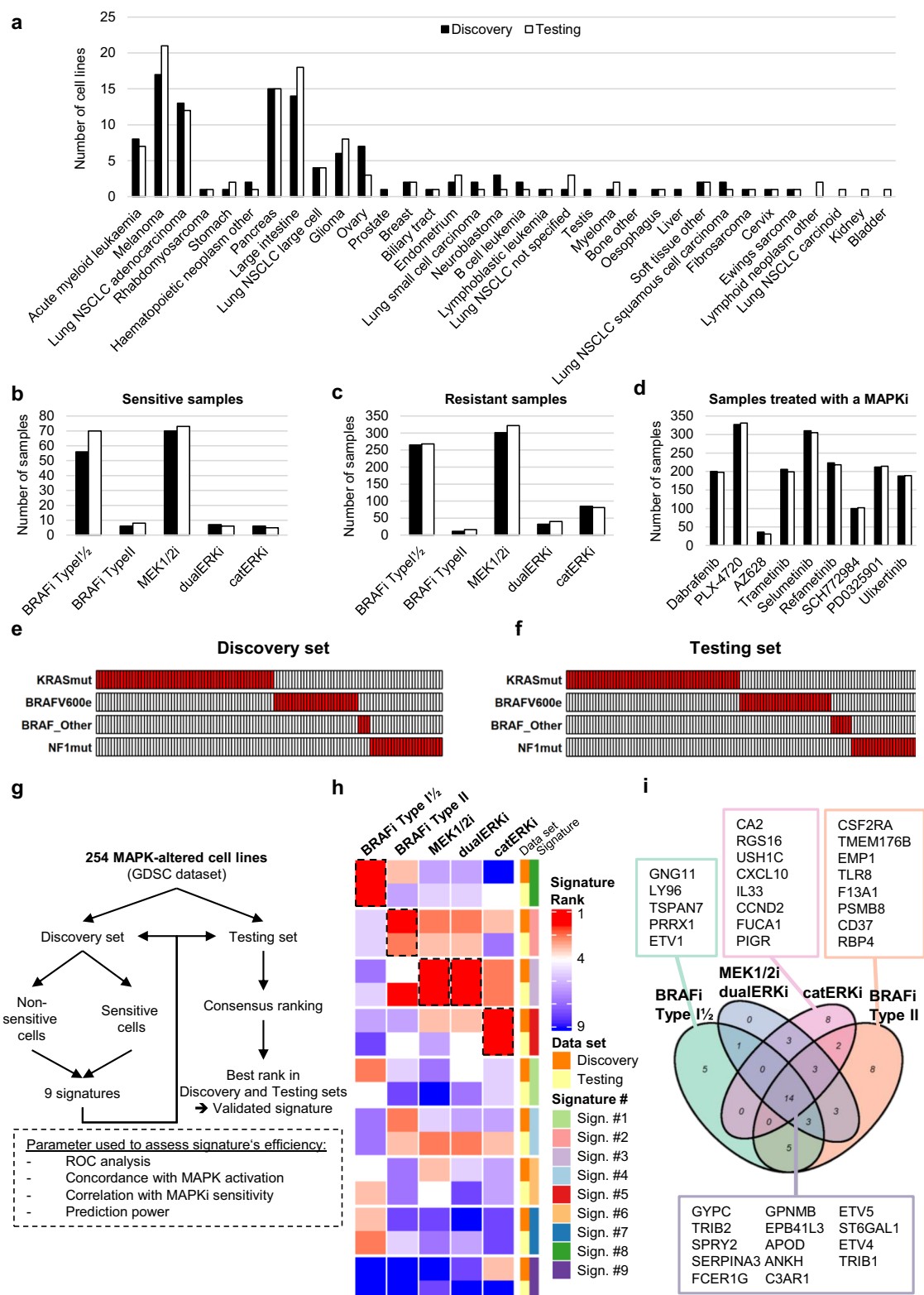

**Fig. 1 | Class-based MAPKi predictive sensitivity signatures validation.** The GDSC dataset was split into a Discovery and Testing set, making sure that both sets had no major differences. To do so, both datasets contain the same proportions of **a** cell lines derived from a given tumor type, **b** sensitive and **c** resistant samples to a given type of MAPKi, **d** samples treated with a given MAPKi, **e**, **f** cell lines with a given MAPK alteration. **g** Simplified overview of the pipeline used to generate the MAPKi sensitivity signatures. **h** Heatmap depicting the final consensus ranking for

all nine signatures based on their ability to best predict sensitivity to a given class of MAPKi. To avoid confusion, signatures were assigned random numbers, but the information from which MAPKi they are derived can be found in Supplementary Data 4. The color in the heatmap represents the rank that the signature reached in the consensus ranking in Discovery and Testing sets. **i** Venn diagram depicting genes overlap between signatures and identification of a potential "Overlap MSS". Source data are provided as a Source Data file.

response; (4) concordance index between scores and measured MAPKi response; and (5) sensitivity prediction accuracy (see "Methods" section). These ssGSEA scores and performance metrics were also measured in the Discovery set that the signatures were derived from to obtain a "best performance" reference to compare with the results obtained in the Testing set (Supplementary Data 3).

For each class of MAPKi (i.e., BRAFi Type I 1/2, BRAFi Type II, MEK1/2i, catERKi, dualERKi), a consensus ranking was used to sort the signatures based on their overall performance (see "Methods" section, and Supplementary Fig. S5a). Consequently, we validated four signatures that had consistent ranks across Discovery and Testing sets based on their ability to predict sensitivity to either class of MAPKi (Fig. 1h). Of note, the same signature reached the top rank for both MEK1/2i and dualERKi. This can be explained by the fact that the samples sensitive to MEK1/2i were also sensitive to the dualERKi (Supplementary Fig. S5b, purple dashed rectangles).

Interestingly, our four signatures showed significant overlap, indicating a possibility for a common signature that would predict sensitivity to MAPKi in general (Fig. 1i). However, when tested for correlation with MAPKi sensitivity, this signature based on the overlapping genes (overlap MSS) did not outperform our individual validated signatures (Supplementary Fig. S6a), indicating that each class of MAPKi seems to require a specific predictive signature. Of note, only 9/55 genes identified in all four MSS signatures were previously described as being involved in MAPKi sensitivity[31,35,36,43,44] (Supplementary Fig. S6b).

In summary, we identified four validated signatures (BRAFi Type I 1/2, BRAFi Type II, catERKi_MEK1/2i and dualERKi) (list of genes available in Supplementary Data 4), which can be used to calculate ssGSEA-derived MAPKi sensitivity scores (MSSs) that correlate with sensitivity to five classes of MAPKi (BRAFi Type I 1/2, BRAFi Type II, MEK1/2i, catERKi and dualERKi) in the GDSC dataset.

In order to illustrate the relevance of our approach, we generated signatures for three drug classes that might be of potential interest for the treatment of pLGG in future studies, i.e., NTRKi, FGFRi, and mTORi. Since the cell lines comprising the dataset were not harboring relevant genetic alterations (NTRK fusions, FGFR fusions or mutations, TSC1/2 mutations), the GDSC dataset was randomly split regardless of the underlying genetic background. Sensitivity data from specific NTRKi (GW441756, AZD1332), FGFRi (AZD4547, PD173074), and mTORi (AZD2014, AZD8055, Temsirolimus, Rapamycin) were used. GSEA was performed using pathway-specific gene signatures ("REACTOME_SIGNALING_BY_NTRKS", "REACTOME_DOWNSTREAM_SIGNALING_OF_ACTIVATED_FGFR", and "HALLMARK_PI3K_AKT_MTOR_SIGNALING", respectively). The same metrics as for the MSS were used to rank the signature, except for the concordance with MAPK pathway activity. Gene signatures topping the ranking in both Discovery and Testing sets were identified, potentially identifying signatures predicting sensitivity to mTORi (Supplementary Fig. S7a), NTRKi (Supplementary Fig. S7b), and FGFRi (Supplementary Fig. S7c). Interestingly, the mTORi signature score was the highest in a set of pediatric sub-ependymal giant cell astrocytomas (SEGA) with *TSC1/2* mutation compared to normal brain tissue from Bongaarts et al. (Supplementary Fig. S7d). The NTRKi signature score was the highest in primary pLGG samples with NTRK fusions from the OPBTA (Supplementary Fig. S7e). Finally, the FGFRi signature was the highest in the subset of pLGG with FGFR alterations compared to pLGGs with other alterations, and two normal tissue samples (Supplementary Fig. S7f), the latter potentially because of high expression of FGFR2 (comprised in the gene signature) in these samples (Supplementary Fig. S7g). Taken together, these data validate our approach to sensitivity signature identification and identify gene signatures (Supplementary Data 4) potentially associated with response to NTRKi, FGFRi and mTORi, which could be further explored in subsequent studies.

## MSSs outperform the MPAS in predicting MAPKi response in an independent PDX cohort

In order to validate the predictivity of our MSSs in an independent and clinically relevant dataset, i.e., including primary material, we used the in vivo dataset from the PDX Pharmacogenomic database (XevaDB, Novartis)[41]. Gene expression data from treatment-naïve patient-derived primary material from carcinomas and melanomas harboring mutually exclusive MAPK alterations only (Fig. 2a) were used to calculate the MSSs. The scores were then compared to changes in tumor volume, i.e., treatment response, in mice transplanted with the corresponding primary material and treated with encorafenib (Fig. 2b), binimetinib (Fig. 2c) or trametinib (Fig. 2d).

Both BRAFi Type I 1/2 and MEK1/2i MSSs showed good concordance (concordance index from 0.62 to 0.66) with treatment response and a statistically significant correlation with treatment response (Fig. 2b–d). When grouped per type of response, the MSSs clearly discriminated between models with tumors responding to the treatment (complete remission: CR, and partial response: PR) and those that showed stable disease (SD) and/or experienced progressive disease (PD) (Fig. 2b–d). However, statistical significance was only reached for the encorafenib-treated cohort (Fig. 2b). In contrast, neither the MPAS (Supplementary Fig. S8a–c) nor our putative overlap-MSS (Supplementary Fig. S8d–f) reached such results in this independent PDX cohort, with lower concordance indices and low correlation coefficients not reaching statistical significance. Finally, a ROC curve following a simple logistic regression for the prediction of encorafenib, binimetinib or trametinib based on the measured BRAFi Type I 1/2 MSS or MEK1/2i MSS showed good efficacy in predicting treatment outcome in the PDXs, with sensitivities and specificities ranging from 67 to 83% and 71 to 91% respectively (Fig. 2e). In each case, our MSSs outperformed the MPAS and the overlap MSS.

None of the BRAFi Type I 1/2 and MEK1/2i MSS correlated with tumor progression in the untreated group, allowing to suggest that the signatures are not affected by a confounding effect driven by the spontaneous evolution of the tumor (Supplementary Fig. S9a). In addition, none of the tested signatures predicted response to chemotherapeutic agents such as 5-FU, gemcitabine, and dacarbazine (Supplementary Fig. S9b–d), indicating specificity toward MAPKi sensitivity rather than a general treatment response.

Taken together, our data demonstrate that our MSSs are able to predict response to MAPKi in vivo specifically, regardless of the natural course of tumor progression, and that they outperform the MPAS to predict sensitivity to BRAFi Type I 1/2 and MEK1/2i.

## MSSs predict MAPKi response in pLGG cell lines in vitro

Since our signatures were derived from and validated in a pan-cancer background, we decided to evaluate them in a pLGG-specific background. We measured the MSSs in a panel of pLGG cell lines[45,46] (*n* = 5) comprising pilocytic astrocytoma (PA) cell lines harboring a *KIAA1549:BRAF* fusion (DKFZ-BT66, DKFZ-BT308, DKFZ-BT317), a PA and a pleomorphic xanthoastrocytoma (PXA) cell line harboring a *BRAF*^V600E mutation (DKFZ-BT314 and BT40, respectively), melanoma cell lines (*n* = 55) with *BRAF* mutations and known high MAPK pathway activity[36] (from GSE7127; used as positive control), and multiple myeloid cell lines (*n* = 11) with MAPK wild-type and known low MAPK pathway activity[36] (from GSE6205; used as negative control). The pLGG cell lines with a *BRAF*^V600E mutation showed BRAFi MSSs at the same level as BRAF-mutated melanoma cells and had significantly higher MSSs compared to the pLGG cell lines with a *KIAA1549:BRAF*-fusion (Fig. 3a and Supplementary Data 5). The catERKi MSS was high in the MAPK wild-type multiple myeloma cell lines, although consistent with a recent study showing the efficacy of the catalytic ERKi LY3214996 in multiple myeloma cell lines, even in the absence of a MAPK alteration (OPM2 cell line)[47].

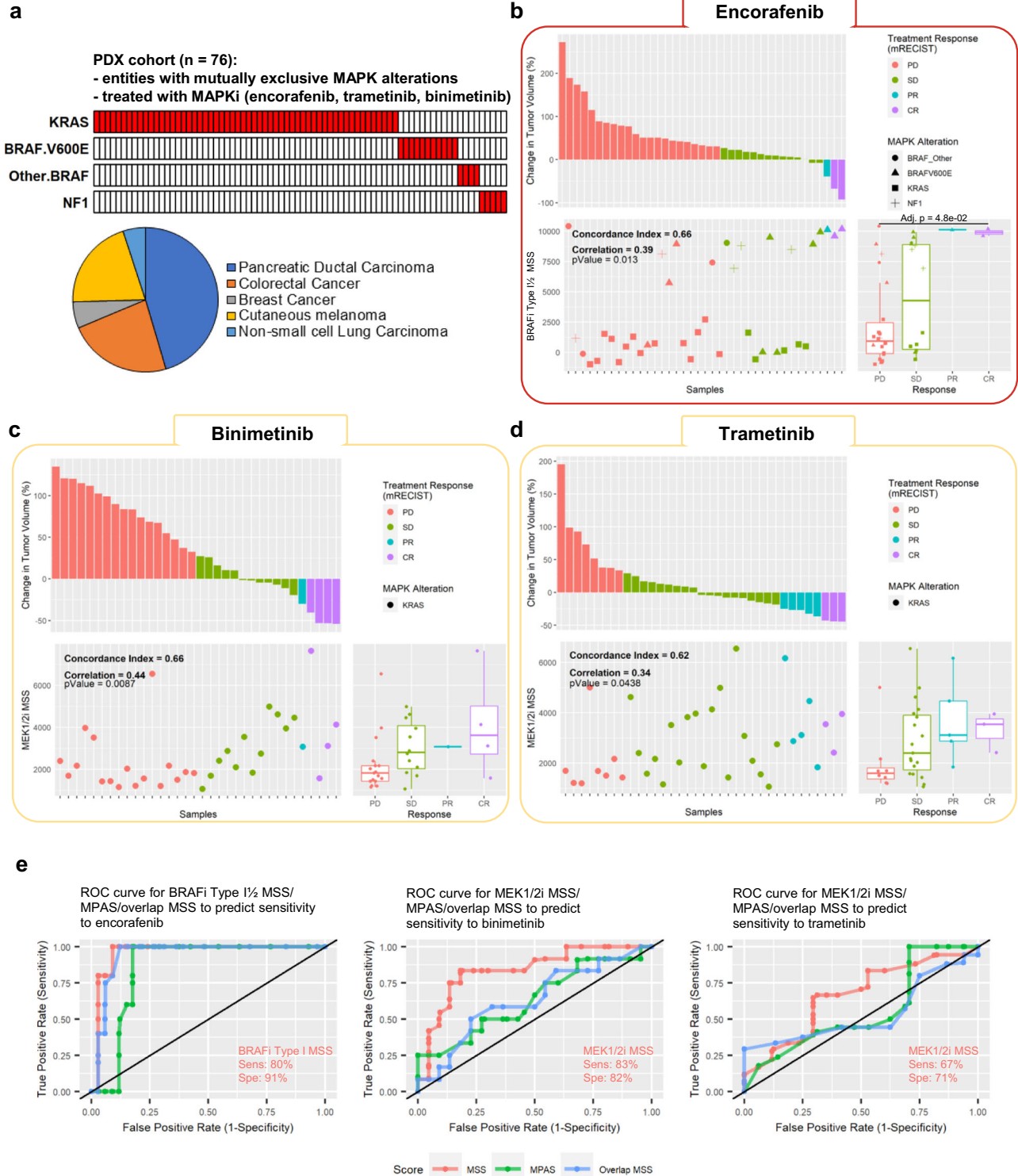

**Fig. 2 | Independent validation of MSSs in a PDX trial cohort.** Gene expression data paired with MAPKi response from the XevaDB PDX cohort (Novartis) was used to validate the signatures. **a** PDXs derived from several tumor entities and with mutually exclusive MAPK alterations were used. Waterfall plots were used to depict MAPKi treatment response (i.e., primary response as described in the original publication; mRECIST criteria) for each sample, and dotplots were used to depict corresponding MAPKi sensitivity scores. Samples were grouped based on treatment response in boxplots. This tryptic analysis was done for PDX treated with **b** the BRAFi type I 1/2 encorafenib, **c** MEK1/2i binimetinib, and **d** trametinib.

Boxplots depict the median, first and third quartiles. Whiskers extend from the hinge to the largest/smallest value no further than 1.5 * IQR from the hinge (where IQR is the interquartile range). Significance was calculated with one-way ANOVA followed by Tukey's 'Honest Significant Difference' test, not significant if not specified. **e** Receiver operating characteristic (ROC) curve for prediction of encorafenib, trametinib or binimetinib based on the measured BRAFi Type I 1/2 MSS or MEK1/2i MSS. Are also indicated sensitivity (sens) and specificity (spe) at best MSS threshold as identified by Youden's J statistics. Source data are provided as a Source Data file.

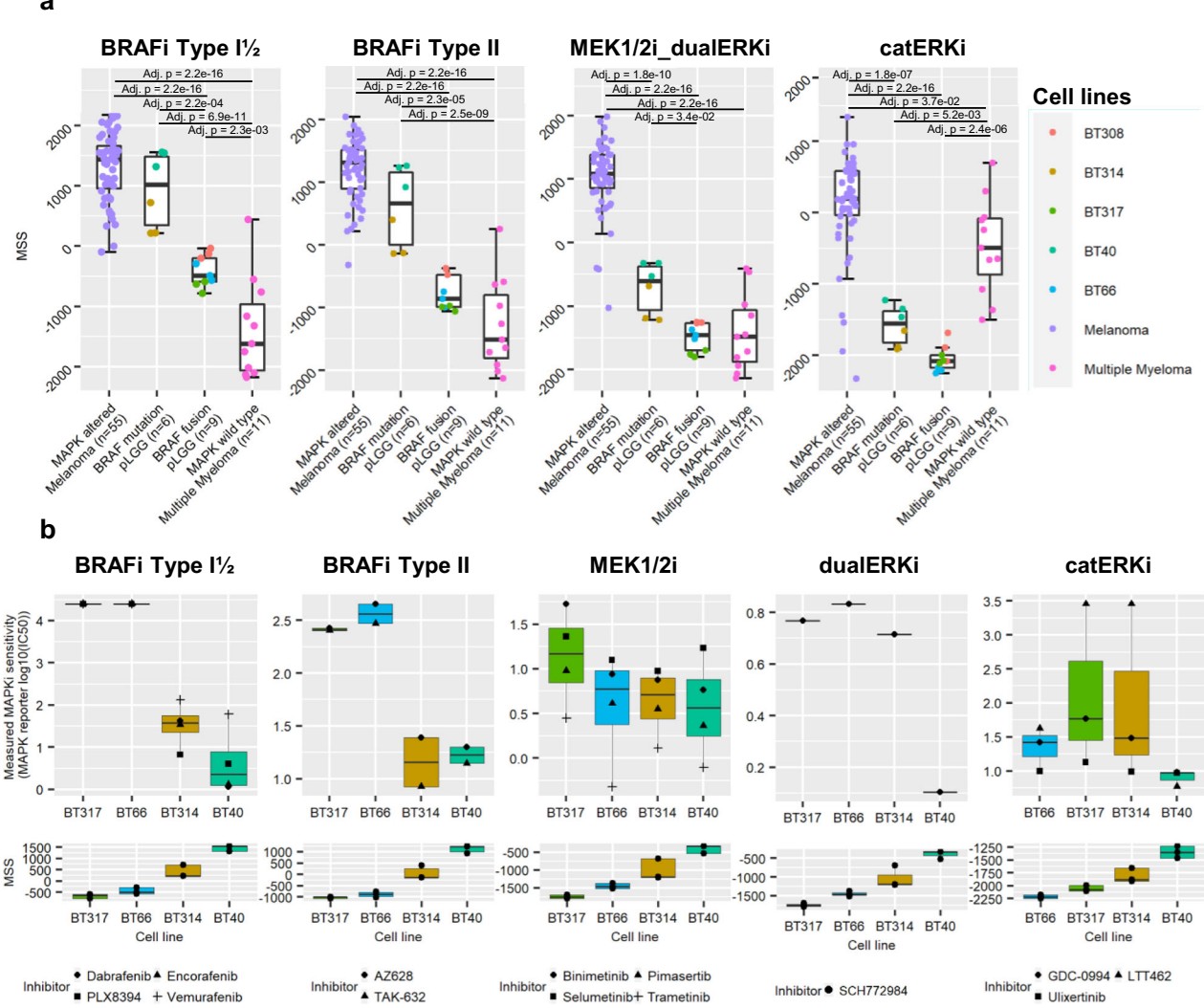

**Fig. 3 | MSSs in pLGG cell lines. a** Scores were measured in a panel of pLGG cell lines with *BRAF*[V600E] mutation (DKFZ-BT314 and BT40, examined over three independent experiments), with *KIAA1549:BRAF* fusion (DKFZ-BT66, DKFZ-BT308 and DKFZ-BT317, examined over three independent experiments), melanoma cell lines with *BRAF* mutations ($n = 55$ biologically independent samples), and multiple myeloma cell lines with wild-type MAPK pathway ($n = 11$ biologically independent samples). Boxplots depict the median, first and third quartiles. Whiskers extend from the hinge to the largest/smallest value no further than 1.5 * IQR from the hinge (where IQR is the interquartile range). Colored dots represent biological triplicates.

One-way ANOVA followed by Tukey's 'Honest Significant Difference' was used to measure significance. Not significant if not specified. **b** Matching analysis was then done between MAPKi sensitivity data measured from a MAPK-reporter assay and the corresponding MSS in the BT40, DKFZ-BT66, DKFZ-BT314 and DKFZ-BT317 cell lines to estimate the reliability of each class-based MSS to predict MAPKi sensitivity. Boxplots depict the median, first and third quartiles. Whiskers extend from the hinge to the largest/smallest value no further than 1.5 * IQR from the hinge (where IQR is the interquartile range). $n = 4$ cell lines examined over three independent experiments. Source data are provided as a Source Data file.

We then sought to determine whether the MSSs can predict sensitivity to MAPKi in these pLGG cell lines. As previously described, metabolic activity cannot be used in the DKFZ-BT pLGG cell lines to assess MAPKi sensitivity[45]. To circumvent this, we previously developed a MAPK pathway activity reporter that allows us to measure MAPK pathway activity response to MAPKi in these models[48]. We performed a MAPKi mini-screen in the DKFZ-BT314 cells (*BRAF*[V600E] mutation) and DKFZ-BT317 cells (*KIAA1549:BRAF* fusion) transduced with this reporter and determined the IC$_{50}$ of several MAPKi belonging to of each class investigated in this study (Supplementary Figs. S10 and S11, respectively). We collected the corresponding IC$_{50}$ in DKFZ-BT66 and BT40 under similar culture and treatment conditions which our group has previously published[48] and pooled the IC$_{50}$s. Comparison of the MAPKi IC$_{50}$ values with the MSSs from BT40, DKFZ-BT66, DKFZ-BT314 and DKFZ-BT317 showed that higher MSSs correlated with lower IC$_{50}$s, indicating higher sensitivity to the

given MAPKi (Fig. 3b). We conclude that the MSSs could predict MAPKi sensitivity in the pLGG cell lines tested in vitro.

## MSSs are elevated in MAPK-driven pLGG and identify subgroups of patients that could benefit from MAPKi therapy

In order to validate the MSSs in primary pLGG, we calculated these scores in pediatric brain tumor samples from the OPBTA dataset[49] and a TCGA-derived dataset comprising RNAseq data from various tumor entities and corresponding normal tissue as control (Supplementary Fig. S12a, b). In the OPBTA dataset, pLGG had the highest MSSs, with a median score significantly higher than the overall dataset's median (Fig. 4a). All other entities with MSSs higher than the overall median either harbored a MAPK pathway alteration (glial neuronal tumors, infantile hemispheric glioma) or had already shown to be targetable by MAPKi (chordoma[50], craniopharyngioma[51], diffuse midline glioma[52], neurocytoma[53] and subependymal giant cell astrocytoma[54]).

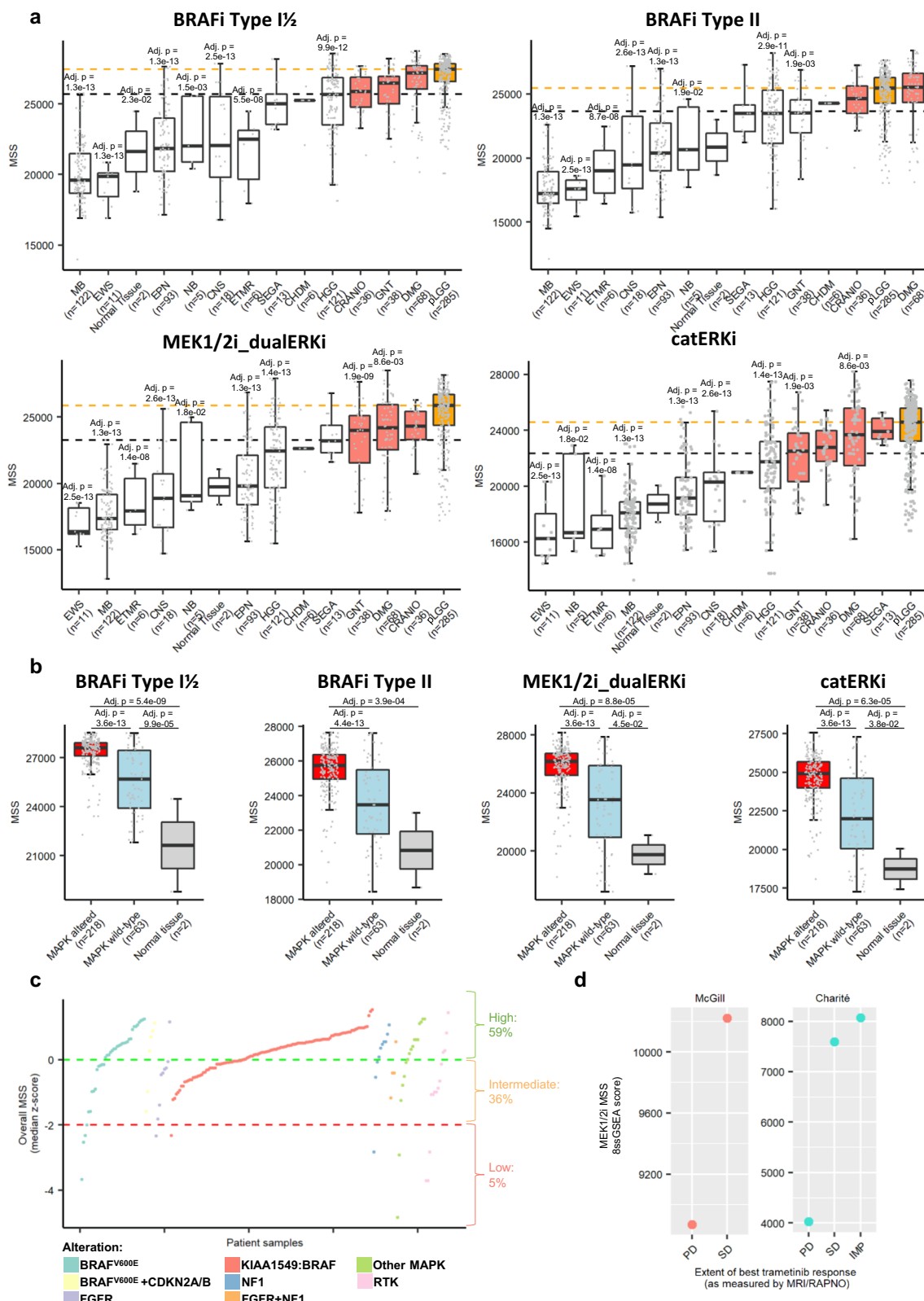

Consistently, in pLGG entities only, the samples harboring a MAPK alteration showed higher MSSs compared to those with an unaltered MAPK pathway and normal tissue (Fig. 4b).

To compare the MSSs to published response rates in clinical trials and to describe the dynamic range of the MSSs across large patient cohorts, we investigated the capability of the overall MSS (median of all four MSSs) to identify potential overall MAPKi-sensitive clusters of

pLGG samples in the OPBTA dataset (Fig. 4c). Samples with an overall MSS above the average ($z$-score > 0) were considered to have a high predicted sensitivity score (potentially predicting PR/CR), samples with an overall MSS significantly lower than the average ($z$-score < −2) were considered to have a low predicted sensitivity (potentially predicting PD), and samples with an in-between overall MSS ($z$-score between 0 and −2) were considered to have an intermediate predicted

**Fig. 4 | MSSs in primary samples from pediatric gliomas.** MAPKi sensitivity scores were measured in the OPBTA dataset. **a** Boxplots depicting the corresponding ssGSEA predictive MAPKi sensitivity score for each brain tumor entity. Orange boxes represent pLGG entities and dashed orange lines pLGG median scores. Salmon boxes represent entities with a median score higher than the overall median (dashed black line). One-way ANOVA followed by Tukey's 'Honest Significant Difference' was used to measure significance. The significance compared to the pLGG group is only depicted. Boxplots depict the median, first and third quartiles. Whiskers extend from the hinge to the largest/smallest value no further than 1.5 * IQR from the hinge (where IQR is the interquartile range). MB medulloblastoma, EWS Ewin Sarcoma, EPN ependymoma, NB neuroblastoma, CNS other CNS embryonal tumor, ETMR embryonal tumor with multilayer rosettes, SEGA Subependymal Giant Cell Astrocytoma, CHDM chordoma, HGG high-grade glioma, CRANIO craniopharyngioma, GNT glial neuronal tumor, DMG diffuse midline glioma, pLGG low-grade glioma. **b** Boxplots focusing on pLGG samples only and showing samples with a detected MAPK alteration vs samples with a wild-type MAPK pathway. Boxplots depict the median, first and third quartiles. Whiskers extend from the hinge to the largest/smallest value no further than 1.5 * IQR from the hinge (where IQR is the interquartile range). One-way ANOVA followed by Tukey's 'Honest Significant Difference' was used to measure significance. Not significant if not specified. **c** Dotplot depicting the overall predicted MAPKi sensitivity (median across all four signatures) z-score for the pLGG samples only, and split based on their molecular alteration. The proportion of samples with a high, intermediate and low sensitivity score is also depicted. **d** MEK1/2i sensitivity score was measured in pLGG-derived primary samples from patients who received trametinib treatment. Gene expression was measured on samples acquired prior to treatment initiation. Since the MSS are not comparable across datasets, the samples were split based on the institute of origin. Source data are provided as a Source Data file.

sensitivity (potentially predicting SD). Hence, we identified sample clusters with several levels of predicted sensitivity (Fig. 4c). The proportions of samples in each sensitivity group matched closely to what has been observed in clinical trials: high sensitivity 59% compared to 33–55% PR/CR in clinical trials, intermediate sensitivity 36% compared to 24–67% SD in clinical trials, and low score 5% compared to 4–28% PD in clinical trials[21–24]. Interestingly, the MPAS, while reflecting MAPK pathway activity, did not efficiently differentiate between high and low sensitive clusters (Supplementary Fig. S13).

We then sought to validate the predictivity of our MSSs in pLGG using sequencing data from MAPKi treatment-naïve primary samples from patients who subsequently received a MAPKi therapy (trametinib) and for which treatment response was available by MRI using the RAPNO cirteria[55] (see Supplementary Data 6). We showed that patients with the best response (SD-IMP) type had a higher MEK1/2i MSS than those with the poorest response (PD) (Fig. 4d). Of note, some patients received other treatments prior to or concurrent with MAPKi therapy. Nonetheless, this could indicate that our signatures might be used to identify patients that could benefit from a MAPKi treatment, but also those that would be resistant to the therapy.

To address the current lack of clinical response data of pLGG patients to MAPKi paired with RNAseq data from baseline, we analyzed a publicly available RNA sequencing dataset from melanoma patients, with mutually exclusive MAPK alterations and which were treated with the BRAFi Type I 1/2 vemurafenib. Notably, the data showed that patients who responded well to the treatment had a higher median BRAFi Type I 1/2 MSS than those who did not respond, and this difference was almost significant despite the small sample size (Supplementary Fig. S14). This finding supports the notion that the MSS might be useful in predicting response to treatment and should be further evaluated and validated for clinical use.

Subsequently, we evaluated all four MSSs and the MPAS in other non-central nervous system (CNS) tumors and their corresponding normal tissue (Supplementary Fig. S15a–e). While the MPAS predicted higher MAPK pathway activity in the MAPK-altered subgroups, not all tumor entities were predicted to be sensitive to MAPKi therapy by the MSSs (Supplementary Fig. S15f). Interestingly, a non-exhaustive literature research showed a correlation between the predictive response to a MAPKi by MSSs and what was observed in clinical trials[56–66]. This underlines the different properties of the MPAS and MSS described in this study, with the MPAS measuring MAPK activation and the MSS potentially predicting sensitivity.

We also investigated whether the MSS could be useful in predicting response to combination treatments, as patients are often treated with a combination of multiple MAPKis. To do so, we first treated two pLGG models (DKFZ-BT314: *BRAF*^V600E mutation; DKFZ-BT317: *KIAA1549:BRAF* fusion) with the first FDA approved systemic therapy for first-line treatment of pediatric patients with pLGG with a *BRAF*^V600E mutation, namely trametinib and dabrafenib. Since dabrafenib is not recommended for the treatment of *KIAA1549:BRAF* fusion

pLGG, we also tested a combination of the MEKi trametinib with the currently clinically investigated BRAFi Type II tovorafenib (DAY101). For both of these BRAFi+MEKi combinations, we could show an inverse correlation between the measured combination log10(IC$_{50}$) and the predicted sensitivity to both combined drugs, i.e., increased MAPKi sensitivity correlated with increased MSS (Supplementary Fig. S16a). Other combinations were assessed in recently published data on combinations of the first-in-class catalytic ERKi (ulixertinib) and several MEKi (trametinib, selumetinib, and binimetinib) in several pLGG models (*BRAF*^V600E mutation and *KIAA1549:BRAF* fusion)[67]. Using these data, we could show the same correlation between increased sensitivity to the combination treatment and increased MSS (Supplementary Fig. S16b). Finally, using data from the XevaDB of in vivo PDX treated with a combination of the MEKi binimetinib and the BRAFi Type I1/2 encorafenib, we showed that the MSS for each drug was positively associated with the response to the combination treatment, i.e., the higher the MSS, the better the PDX responded (Supplementary Fig. S16c), with the best responses in samples with high MSSs for both drugs. In summary, these findings suggest that the MSS might be useful in predicting response to combination treatments as well and may be a valuable tool for clinical combination therapies.

We conclude that the determination of MSS is applicable in primary pLGG and that the MSSs can differentiate tumors with higher and lower sensitivity scores within entities of the same driving alteration. Prospective validation of the MSSs in clinical trials, including larger patient cohorts treated with the respective MAPKi, is needed.

## MSSs identify immune infiltration as a key factor in the predicted response to MAPKi therapy

We then investigated what factors were significantly associated with our MSSs (continuous outcome variable) in the pLGG samples from the OPBTA, such as age at diagnosis, tumor site, sex and molecular subtype (predictors). We also included the prediction of immune and stromal infiltration in order to evaluate whether the tumor microenvironment could have an impact on the predicted MAPKi sensitivity using the ESTIMATE signatures[68]. Multiple linear regression analysis showed that the predicted immune infiltration was highly significantly associated with MSSs (Fig. 5a and Supplementary Data 7), indicating that the microenvironment compartment could play a role in the degree of sensitivity to MAPKi in pLGG.

The MSSs showed a strong correlation with the ESTIMATE score (aggregation of both immune and stromal scores, reflecting the degree of infiltration) in the pLGG samples from the OPBTA cohort (Fig. 5b), and in particular with the estimated immune infiltration score (Fig. 5c). In contrast, the correlation with the estimated stromal infiltration was weaker (Supplementary Fig. S17). Consistently, the primary pLGG samples that showed the highest overall predicted sensitivity to MAPKi (Fig. 4c) were the samples with the highest predicted proportion of immune infiltration, while the samples with the lowest relative predicted sensitivity had the lowest proportion of predicted immune

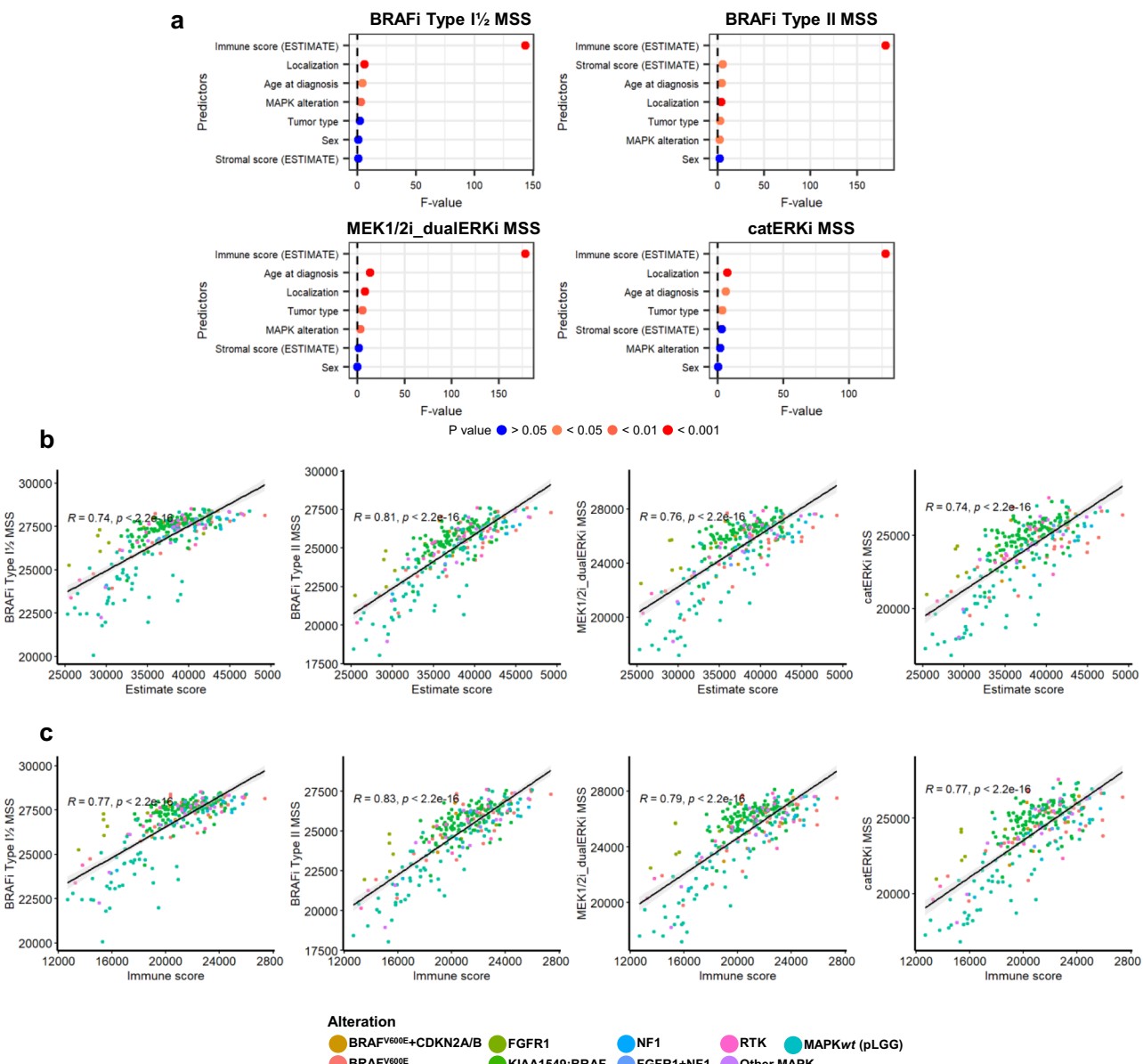

**Fig. 5 | MSSs identify the immune infiltration as a key player in MAPKi response in pLGG.** The ESTIMATE algorithm was used to approximate the proportion of infiltrating cells in the pLGG samples from the OPBTA dataset. **a** Dotplot depicting the F-values obtained after multiple linear regression analysis followed by one-way ANOVA to assess what coefficients are significantly associated with MSS. **b** Correlation between MAPKi sensitivity scores and ESTIMATE scores are depicted, along with Pearson's coefficient of correlation, the corresponding *p*-value (two-tailed *t*-test), and the 95% confidence interval (error band). **c** The correlation between the MAPKi sensitivity score and the immune signature score is also depicted. Significance was calculated with a two-tailed *t*-test. Source data are provided as a Source Data file.

infiltration (Supplementary Fig. S18). Conversely, the MPAS did not correlate well with the total ESTIMATE or immune-specific scores (Supplementary Fig. S19).

To validate this, we measured the MSSs in single-cell RNA sequencing (scRNA-seq) data from 6 pLGG primary samples (Supplementary Data 6). The analysis of key cell population markers (Fig. 6a and Supplementary Fig. S20) allowed us to differentiate between glial-type (*n* = 5) and neuronal-type (*n* = 1) tumors, as well as lymphoid and myeloid cell populations. As expected, the ESTIMATE immune score was exclusively limited to the lymphoid and myeloid compartments (Fig. 6b). Regarding the different MAPK pathway scores, MPAS was significantly enriched in the glial tumor-type compartment, while the four MSSs were elevated in both the glial tumor and myeloid compartments, with the latter showing higher values in microglia than in infiltrating macrophages. The increased MSS in the microglia

compartment was validated in an independent pLGG scRNA-seq data[47] (Supplementary Fig. S21).

Interestingly, the predicted immune infiltration (ESTIMATE immune score) did not correlate with the best MAPKi response type (Supplementary Fig. S22). Overall, the samples with a higher predicted immune infiltration were those with a better outcome, but the relationship was not linear. This suggests that the immune infiltration may play a role in the treatment response but does not overrule the contribution played by the tumor compartment in MAPKi response, as suggested by Fig. 6b, where the MSS is found high in both cell populations.

These data suggest that the microenvironment, and particularly tumor-associated microglia, could be involved, along with the tumor compartment, in the degree of sensitivity toward MAPKi, and might represent a target of MAPKi in pLGG.

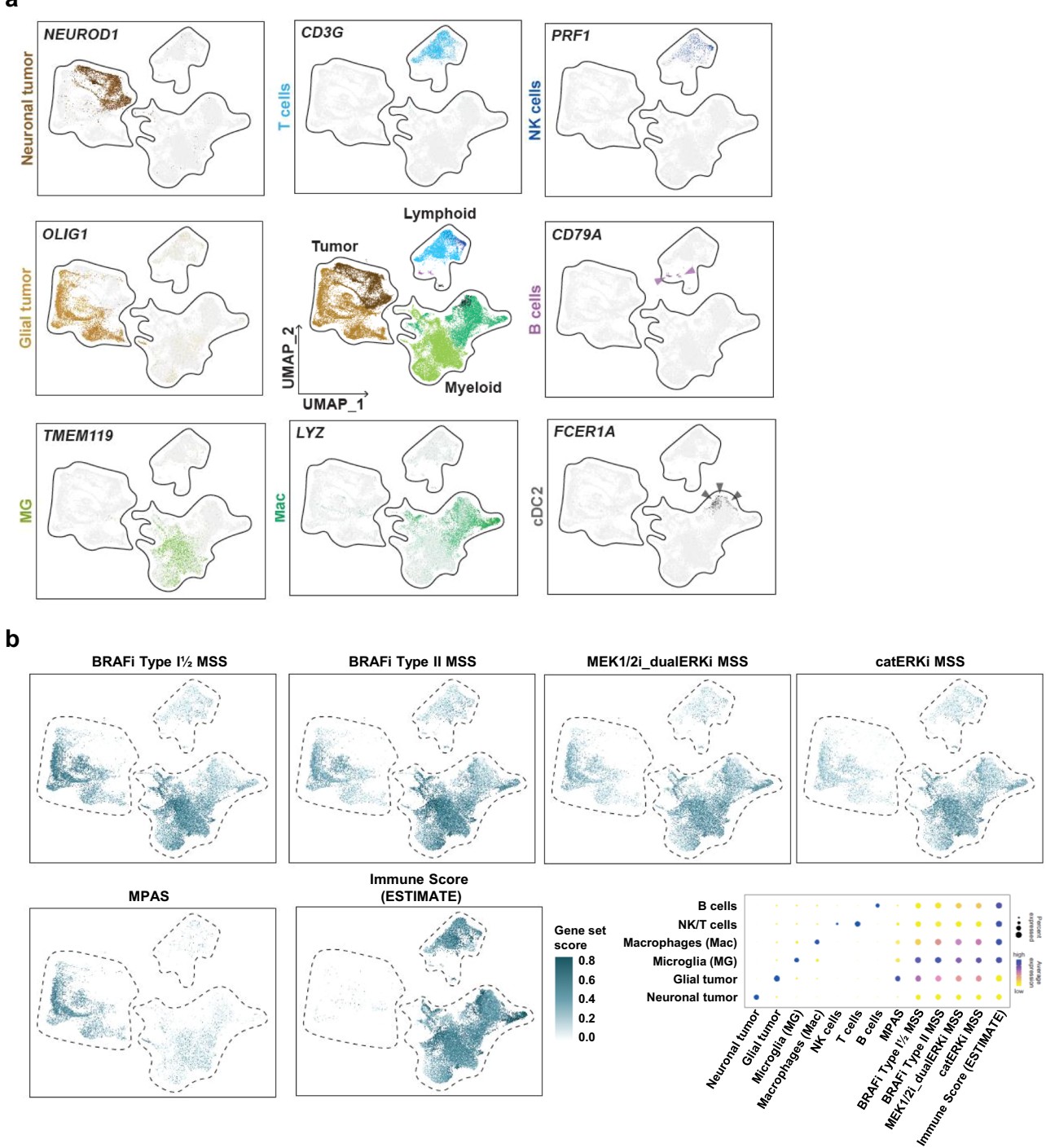

**Fig. 6 | MSSs identify microglia as a key player in MAPKi response in pLGG.** Cell marker expression and MSSs evaluation in a cohort of six pLGG primary samples. **a** UMAPs depicting the key cell population markers for the main cell populations identified in the pLGG samples. **b** UMAPs depicting the MSS, MPAS and immune score (ESTIMATE) in the different clusters. A dotplot summarizing the signature scores in each cluster is also depicted. Source data are provided as a Source Data file.

## MSSs are not confounded and are elevated in tumor-associated microglia

Since the MSSs correlated with the ESTIMATE scores, we investigated the overlap between these signatures to rule out potential biases. We found that only a small number of genes from our signatures overlapped with the immune and stromal signatures (5 and 2 genes, respectively), indicating that the correlation is not biased because of gene overlap (Fig. 7a).

A recent study highlighted the potential for misinterpretation of gene expression-derived signatures due to possible differential gene expression across the tumor epithelium and microenvironmental cells[69]. We tested the extent to which our gene signatures are confounded by the microenvironment transcriptome using the ConfoundR platform. We compared our signatures' enrichment in the stroma compared to tumor epithelium and then within cell populations in the stroma only, using datasets derived from colon, breast,

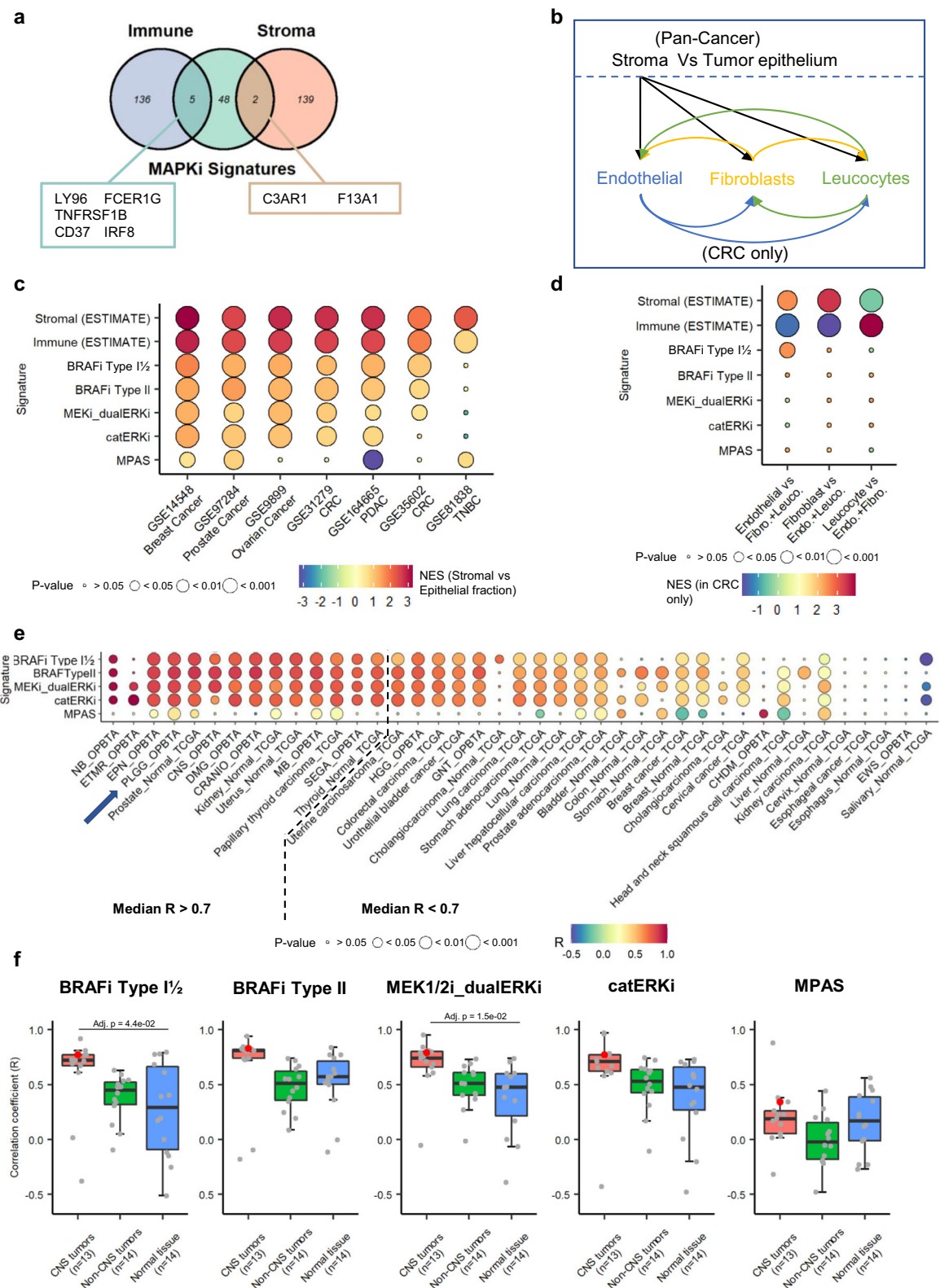

pancreas, ovarian and prostate cancers (Fig. 7b). Our signatures were found to be only moderately enriched (NES < 2) in the stroma, compared to both immune and stromal signatures, here used as positive controls (Fig. 7c). As expected, the stromal signature was significantly enriched in fibroblasts and endothelial cells. In contrast, the immune signature was significantly enriched in the leukocyte population. Conversely, our signatures were not significantly enriched in any of these microenvironmental populations (Fig. 7d). Taken together, these

data indicate that our signatures are not confounded by micro-environmental genes.

Since the ConfoundR platform uses data from non-CNS tumors and therefore lacks data on microglia, we decided to assess the relative expression level of the genes comprised in our signatures in a published scRNA-seq dataset with non-tumor-associated microglia (from Alzheimer's disease or epilepsy from young, middle-aged and aged patients)[70] (Supplementary Fig. S23). MSS genes were expressed in the

Fig. 7 | MSSs are not confounded by gene signature overlap, microenviron-mental transcriptome, or general bias. a Venn diagram depicting genes overlap between an aggregation of our signatures and the immune and stromal signature from the ESTIMATE. In the box are indicated the overlapping genes. b Schematic presenting the comparison studied via GSEA with ConfoundR to test for micro-environmental confounding effect. c Dotplot summarizing the indicated sig-natures' normalized enrichment scores (NES) when comparing stromal vs. epithelial cell populations in six tumor types. Nominal $p$-value from the GSEA output is used. d Dotplot summarizing the indicated signatures' NES when com-paring endothelial, fibroblast or leukocyte populations to the remaining cell populations. Enrichment is depicted by warm colors, while the contrary is depicted by cold colors. Dot size depicts the respective nominal $p$-value from the GSEA output. e Dotplot summarizing the correlation coefficient (R) when comparing the indicated signature to the predicted immune infiltration (from ESTIMATE) in the OPBTA and TCGA datasets. The entities were split based on their median R across all MAPKi sensitivity signatures (i.e., excluding the MPAS). $R > 0.7$ was considered biologically significant. Dot size depicts the respective $p$-value (two-tailed $t$-test). The arrow points at the pLGG samples. f Boxplot grouping the R coefficient by entity type (normal vs tumor) and by localization (CNS vs non-CNS). Each dot represents the coefficient of correlation between MSSs and predicted immune infiltration for $n = 13$ biologically independent CNS tumor types, $n = 14$ biologically independent non-CNS tumor types and $n = 14$ biologically independent normal tissue type. Boxplots depict the median, first and third quartiles. Whiskers extend from the hinge to the largest/smallest value no further than 1.5 * IQR from the hinge (where IQR is the interquartile range). One-way ANOVA followed by Tukey's 'Honest Significant Difference' was used to measure significance. Not significant if not specified. Red dots depict the pLGG entities. CRC colorectal carcinoma, PDAC pancreatic ductal adenocarcinoma, TNBC triple-negative breast cancer. Source data are provided as a Source Data file.

same proportions across all immune and GFAP+ clusters, with overall low expression levels (log(CPM + 1) ≤ 2). Only 2/55 (4%) genes had a high average expression (log(CPM + 1) > 2) (*SPP1* and *FCER1G*) in the microglia clusters, of which one was expressed in only 2/4 MSS signa-tures (*SPP1*), and the other also had an elevated average expression in the monocyte cluster, while our MSSs were specifically elevated in microglia in pLGG. The MPAS genes were also lowly expressed with similar expression proportions across clusters. In contrast, the genes comprising the immune score had similar percentages of expression in the immune clusters (microglia, monocytes, B and T cells) only, with 23/200 (12%) genes highly expressed (log(CPM + 1) >2) in at least one of these clusters. This highlights that the high predicted sensitivity toward MAPKi in microglia might be specific to tumor-activated microglia and not to healthy microglia.

Of note, PDX-derived sensitivity signature analysis did not yield signatures superior to our MSS (see Supplementary Materials and Supplementary Fig. S24).

We finally tested the correlation between the MSS and the pre-dicted immune infiltration in other tumors and tissues from the OPBTA and the TCGA (Fig. 7e). The correlation coefficient between MSSs and the immune score (ESTIMATE) tended to be the highest in central nervous system tumors compared to non-CNS tumors and normal tissues (Fig. 7f), in line with our hypothesis that, among all immune cells involved in the tumor microenvironment, the MSSs are elevated in the microglia compartment in CNS tumors. Of note, this does not rule out a possible role played by tumor-associated macrophages (TAMs), which also present high MSS (albeit lower compared to microglia) in our pLGG scRNAseq dataset (Fig. 6b). Since the difference between CNS and non-CNS tumor doesn't reach significance, it cannot be excluded that monocyte-derived TAMs may contribute to MAPKi response, especially in non-CNS tumors.

Taken together, these data indicate that our MSSs are not biased by genes specifically expressed in stromal/immune cells, and in parti-cular not from healthy microglia, and seem to be specifically elevated in tumor-associated microglia in pLGG and potentially other CNS entities.

## pLGG samples with low MSSs are associated with a neuronal phenotype

We finally investigated whether we could find potential markers/tar-getable molecular mechanisms for the pLGG patients with low MSSs (cluster intermediate and low) that would represent patients with stable or progressive disease (Fig. 4c). Petralia et al. recently demon-strated that pLGG samples could be split into a *Hot* and *a Neuronal* cluster in a mutually exclusive manner. Since our MSSs positively correlated with the proportion of immune infiltration and were the highest in microglia, i.e., in *Hot* pLGG samples, we hypothesized that pLGG samples with low MSSs might be pLGG with neuronal features[71]. To test this, we used a neuronal signature from a scRNA-seq study on

glioblastoma[72] to measure a neuronal score reflecting the proportion of neuronal infiltration. We could show a significant negative correla-tion between the MSSs and the neuronal score in the pLGG samples from the OPBTA (Fig. 8a). The samples from the MSS low clusters had a neuronal score significantly higher than the samples with a high/intermediate MSS (Fig. 8b), suggesting a lower degree of sensitivity for pLGG with neuronal features. In order to control for possible con-founding factors where our MSSs would always positively correlate with the immune score and negatively correlate with the neuronal score, we calculated the correlation between the MSS and the immune/neuronal score in the OPBTA dataset (Supplementary Fig. S25). While the pLGG group had the strongest positive and negative correlation with the immune and neuronal scores, respectively, the diffuse midline gliomas had MSSs that positively correlated with both immune and neuronal scores, hence ruling out confounding bias. We then validated this in our independent pLGG scRNA-seq dataset (Fig. 8c). While the pLGG samples with glial features and high proportion of microglia infiltration had high MSSs in both tumor and microglia compartments (PA1, PA3–5, PMA1), the pLGG sample with neuronal features (PA2) and almost no microglia infiltration had MSSs close to 0. Taken together, these data suggest that pLGG with neuronal features might be at higher risk of not responding to MAPKi compared to pLGG with microglia infiltration.

## Discussion

In the present study, we observed that MAPK pathway activity, as measured by the MPAS, is not an accurate enough surrogate marker to predict responsiveness to any type of MAPKi. We showed that MAPK pathway activity prediction did not correlate well with actual response to MAPKi other than MEKi in the GDSC dataset, in accordance with the fact that the MPAS genes were validated in models treated with MEKi trametinib and cobimetinib[36]. In addition, the MPAS poorly predicted MAPKi response in the XevaDB PDX dataset.

We therefore generated gene expression-based MAPKi sensitivity scores derived from in vitro drug sensitivity from a cohort of more than 200 MAPK-altered cancer cell lines (GDSC dataset). Many studies aiming at identifying gene signatures use unsupervised approaches, such as classifiers using elastic net logistic regression (E-net). Such classifiers have proven useful in generating cell type-specific gene signatures[73]. In our case, the risk of using such unsupervised methods was to select genes that would not only differentiate between MAPKi-sensitive and non-sensitive cells but genes that would also relate to the tissue of origin the cell lines were derived from. To avoid such gene selection bias, we decided to use a biologically supervised approach, focusing on MAPK pathway-related genes only (200 most stable genes comprising the "Hallmark_KRAS_Signaling_UP" signature from the Broad Institute). This rationale was strengthened by the work of Wagle et al., which showed that a biology-driven gene selection performed at least as good as the E-net model[36]. A common pitfall of such supervised

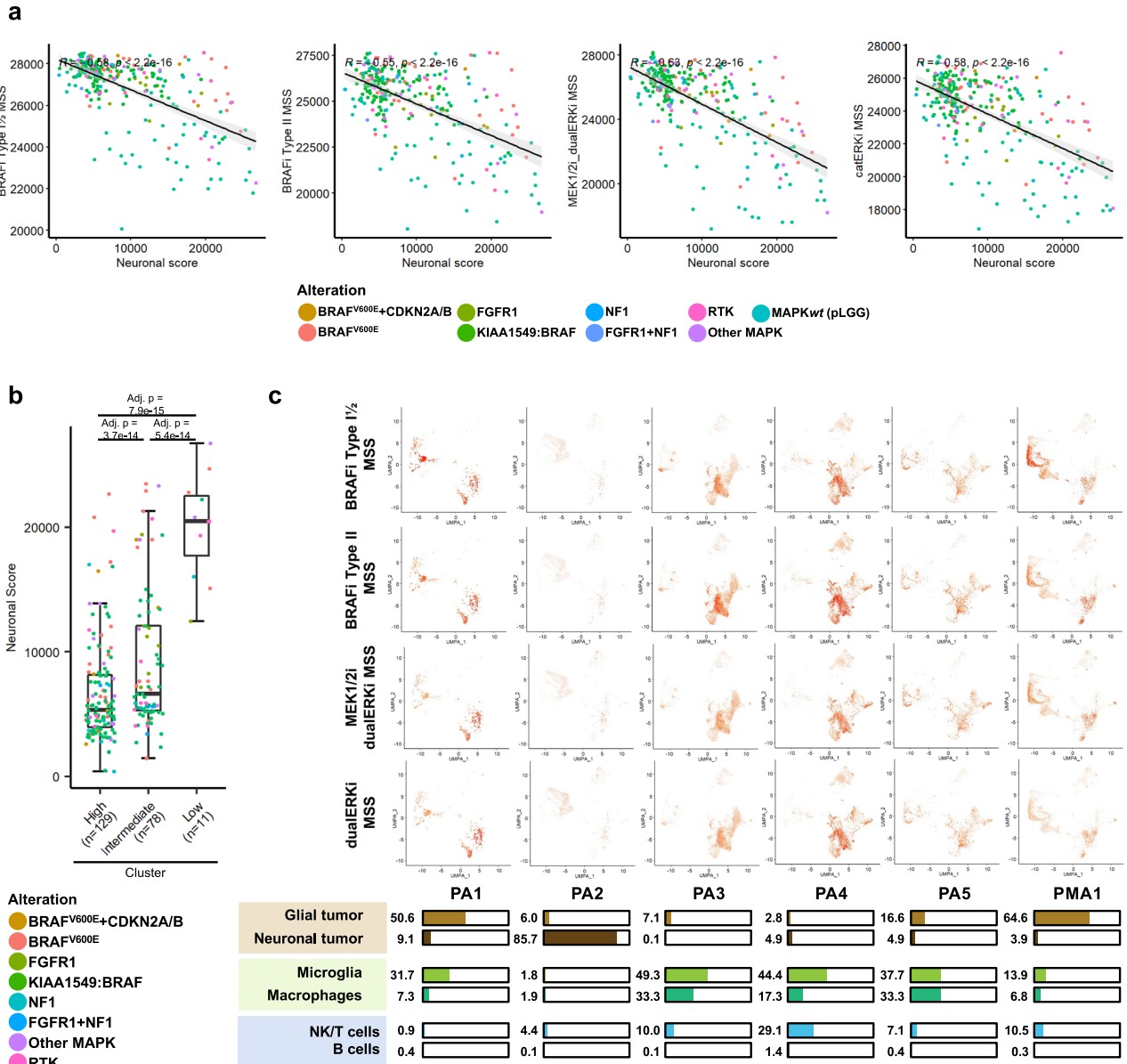

**Fig. 8 | Characterization of pLGG samples with low/intermediate MSSs.**
**a** Correlation between MSSs and the neuronal score is depicted, along with Pearson's coefficient of correlation, the corresponding *p*-value (two-tailed *t*-test), and the 95% confidence interval (error band). **b** Boxplots depicting raw ssGSEA scores for the neuronal score in the clusters from Fig. 4c from the OPBTA cohort. Dots are colored based on the detected driving MAPK alteration. Data from *n* = 218 biologically independent samples were used. Boxplots depict the median, first and third quartiles. Whiskers extend from the hinge to the largest/smallest value no further than 1.5 * IQR from the hinge (where IQR is the interquartile range). One-way ANOVA followed by Tukey's 'Honest Significant Difference' was used to measure significance. **c** UMAPs depicting the MSS in each individual pLGG sample from our scRNA-seq dataset. The relative proportion of each cell population in each pLGG sample is also shown. PA pilocytic astrocytoma, PMA pilomyxoid astrocytoma. Source data are provided as a Source Data file.

prediction, however, is the misuse of split datasets, where the selection of genes is made on the full dataset[74]. We made sure the split of the GDSC dataset was proportional and that the signatures were generated using the Discovery set only. The second common pitfall of using such split datasets is the risk of confounding factors hidden in the dataset[75]. To avoid this, we used an independent PDX dataset to validate the robustness of our MSSs. Another shortcoming when using gene expression-derived datasets has recently been discussed concerning gene-expression signatures derived from bulk tumors being biased by the expression of stromal and microenvironment cells[69]. To control for this bias, we used the published ConfoundR platform that showed that none of our signatures were confounded by gene expression from the microenvironment and stromal cells. This was of particular importance since the Hallmark_KRAS_Signaling_UP has been shown to be

confounded by gene expression from the stroma (using the ESTIMATE stromal score) in all five tumor entities where this was investigated[69]. It appears that our selection of genes specifically upregulated in MAPKi-sensitive tumor cell lines was enough to correct this bias, as our signatures poorly correlate with the ESTIMATE stromal score. All this allowed us to generate biologically robust signatures used to calculate their corresponding MSSs.

We were able to apply this approach to other pLGG-relevant drugs (e.g., NTRKi, FGFRi, and mTORi), generating signatures that could be further explored in future studies. This highlights the relevance of such a biology-driven approach to identify new drug-sensitivity signatures from large pharmaco-genomics databases. A more comprehensive analysis of the whole GDSC dataset, which includes hundreds of compounds, is warranted, as it could lead to deeper discoveries and

further validate this approach. While our approach didn't use nor generate new tools to measure sensitivity score per se, the combination of the methods used allowed us to generate specific and unbiased gene signatures that may capture MAPKi sensitivity in pLGG tumors. In particular, our signatures were derived focusing on MAPK-related genes only (Hallmark_Kras_UP signature) in order to avoid the inclusion of any genes that would be specific to the tissue the tumor cells were derived from. By doing so, we aimed to generate MAPKi sensitivity signatures that would be free of any tissue/cell type-specific genes (as confirmed by our confoundR analysis and PDX-derived MSS analysis) to make the signatures virtually applicable to any cell type.

Our four signatures showed only a small overlap with already published signatures related to MEKi and BRAFi response, as well as MAPK activation (9/55 genes comprised in all four signatures). Interestingly, the overlap across all signatures (overlap MSS) did not perform as well as the single signatures to predict sensitivity to any MAPKi. This allows to suggest that each class of MAPKi, despite targeting the same pathway, probably has a distinct output requiring specific predictive signatures.

Our MSSs showed consistent results in the pLGG background, showing a clear dependency toward MAPK alterations, and accurately predicted MAPKi sensitivity in a set of three pLGG cell lines in vitro. These signatures, derived from in vitro cell lines, precisely recapitulated pLGG biology in primary tumor samples. Indeed, the MAPKi sensitivity signatures showed the highest scores in MAPK-altered gliomas, and more particularly in low-grade gliomas/astrocytomas. The MSSs also captured the proportion of responding patients observed in recent clinical studies[21–23], with scores indicating a proportion of predicted responsive patients of around 59% and nonresponsive patients of about 4% in a cohort of primary pLGG samples. We demonstrated that the MEK1/2i MSS correctly predicted treatment response in five primary pLGG samples, with increasing MSS correlating with a better response. Furthermore, we were able to demonstrate a similar pattern in an independent melanoma dataset in response to the BRAFi Type I 1/2 vemurafenib. This indicates that, beyond pLGG, the MSSs described here could potentially also be applied to predict MAPKi sensitivity in melanoma patients with mutually exclusive MAPK alterations. Although these data warrant further validation in larger datasets, this clearly highlights the potential for such scores to be further explored and validated in future clinical trials. It is worth noting that in the context of combination therapy, our MSS was also positively associated with the response to several types of MAPKi in vitro and in PDXs in vivo. It appeared the best responders were the samples with the highest MSS for both drugs (as opposed to high MSS for only one drug), which may suggest that patients with high MSS for two drugs might be more likely to benefit from combination therapy. This observation will certainly need further validation in upcoming clinical trials.

To date, several studies have shown the potential of such gene signatures derived from preclinical studies and large datasets such as the GDSC or TCGA[76–80]. However, translating the signatures' output into normalized scores, applicable to patient-derived gene expression data from any cohort, remains a challenge for their translation into clinical studies. For instance, the MPAS uses the sum of gene expression's z-score, hence normalizing the scores across the whole dataset, making its value vary when one or several samples are added or removed. To circumvent this, we preferred the ssGSEA approach[81], allowing us to generate an enrichment score per sample independent of the cohort composition and variability in the number of samples. However, one pre-requisite to allow comparison is that every sample has to be comparable, i.e., went through the same sample processing and data processing pipeline, in order to obtain meaningful enrichment scores, making scores comparable across several datasets/studies a challenge. Another challenge is the comparison of scores among themselves, as it is at the moment not possible to compare ssGSEA

scores (i.e., MSSs) with each other, which would allow to estimate whether one MSS is higher or lower than another. Some approaches could be envisioned to circumvent these pitfalls, such as a resampling procedure to generate null distributions for each of the MSSs[82], which will need further validation in prospective studies.

We observed that the MSSs strongly correlated with the ESTIMATE score, particularly its immune signature in primary pLGG samples. On a single-cell level, the MSSs were the highest in the microglia compartment. This was unexpected, considering that our signatures were exclusively derived from gene expression from tumor cell lines and did not show indications of stromal-related gene expression bias. Several studies have shown an interconnection between tumor microenvironment and resistance to MAPKi in certain tumor entities, mainly via dysregulation of macrophages[83] and CD8 + T lymphocyte[84] activity. In addition, the MAPK pathway is known to be active, in particular in macrophages/microglia cells, where it is involved in microglia/macrophage polarization, metabolism, and pro-inflammatory activities[85–87]. A recent study also described the dependency of migrating glial progenitors toward MAPK pathway activity in the development of NF1-associated optic pathway gliomas, where MAPK inhibition was capable of preventing the expansion of such glial progenitors[88]. This indicates that the MAPK pathway is a key pathway in microenvironmental cells, particularly in microglia cell biology. Since we showed that healthy human microglia and infiltrating macrophages had low MSSs, it could be that microglia become sensitive to MAPKi only following interaction with tumor cells, either via direct contact or stimulation by secreted factors, such as the senescence-associated secretory phenotype (SASP) factors[89]. This predicted MAPKi sensitivity of tumor-associated microglia in pLGG will need further validation. In vitro validation might represent a challenge, however, considering that all the current microglia cell models present limitations[90], the most important being their immortalization with the SV40-largeT antigen, preventing any TP53-related apoptosis, as already observed in SV40-transformed pLGG models[45]. scRNA sequencing on short-term cultured primary samples treated with MAPKi, or pLGG organoids might represent an alternative to investigate this further. If confirmed, and considering the importance of microglia in the support of glioma development[91], this finding could change the way we see pLGG treatment with MAPKi, and, more generally, in the way we consider pLGG biology: patients treated with MAPKi therapy show signs of tumor volume reduction; however, the main cell populations targeted are only hypothesized as being exclusively pLGG tumor cells. In addition, the fast rebound often observed upon treatment cessation is a counter-intuitive phenomenon to happen in tumors that are thought to be mainly in a senescent status. Based on our results, it is conceivable to imagine that pLGG is partly driven by immune cells, which are recruited and supported by senescent tumor cells[91]. If microglia are at least as sensitive to MAPKi as the glial tumor cells, such therapies might induce a reduction of the proportion of the microglia compartment (which can represent up to 30% of a pLGG tumor[92]) as at least a part of the reduction of tumor volume. Upon MAPKi withdrawal, microglia might be able to be quickly recruited by the remaining surviving tumor cells via the production of SASP factors, at least partly explaining the fast tumor rebound. This hypothesis is worth further investigation in the future.

The MAPKi sensitivity prediction in the immune compartment also underlined a limitation of one of our signatures: the BRAFi type I 1/2 sensitivity signature predicted sensitivity in the normal microglia compartment (i.e., BRAF wt cells). It is well known that such BRAFi generally induces paradoxical activation of the MAPK pathway in non-BRAF-altered cells[93,94]. Therefore, the immune cell population should not be detected as a putative sensitive population. This false positivity probably came from the fact that, by design, the signature relates to MAPK pathway activity, which is increased in both tumor and microglia cells (although to a lower extent), while the paradoxical activation

induced by BRAFi Type I 1/2 is purely intrinsic to the drug's mode of action[10]. Therefore, a cell population with an activated MAPK pathway could be wrongly predicted as being sensitive to a BRAFi Type I 1/2. However, this signature still accurately predicted the degree of sensitivity to a BRAFi type I 1/2 in our GDSC-derived Testing set and the independent PDX dataset, advocating that this signature could still be relevant under the condition that the type of genetic MAPK alteration is known before treatment initiation[93].

Our signatures also allowed us to investigate potential molecular and cellular mechanisms driving pLGG tumors with low sensitivity toward MAPKi. We found that pLGG with neuronal features seems to be less sensitive to MAPKi, as estimated with our MSSs. Petralia et al. described a dichotomous classification of pLGGs in either Hot tumors with a high degree of immune cell infiltration or Neuronal with a low degree of immune cell infiltration[71]. In line with this observation, the correlation of high MSS with immune cell infiltration in the Hot tumors was inversed in the Neuronal tumors in our dataset. However, other factors than just plain low immune cell infiltration may play a role in decreased MAPKi sensitivity in Neuronal-type pLGGs. Interestingly, it has been shown in adult low- and high-grade gliomas that neurons could form synapses with glioma cells, hence forming so-called neuronal-glioma synapses (NGS)[95,96]. These NSG have been shown to promote glioma cell progression and survival through the activation of several pathways, such as the Akt/mTOR pathway and the MAPK pathway, which are mediated by N-methyl-D-aspartate (*a.k.a.* NMDA) and amino-3-hydroxy-5-methyl-4-isoaxazolepropionate acid (*a.k.a.* AMPA) receptors[97]. If validated in a larger dataset, this could potentially pave the way for a new type of therapy for the treatment of pLGG, where MAPKi could be used in combination with neurotransmitter inhibitor, which could potentiate and increase the effects of MAPKi in low responders/resistant pLGG patients.

Larger gene expression datasets derived from patients that received a MAPKi treatment, coupled with their respective response to the treatment (such as in the upcoming LOGGIC Trial), are now urgently needed to identify and validate sensitivity score thresholds to assess the applicability of our MSS as stratification tools for clinicians. Such large datasets will also allow further investigation of the involvement of the microglia in treatment response and the relationship between pLGG with neuronal features and MAPKi resistance. This will ultimately allow a better understanding of pLGG biology and the identification of new treatment modalities that could improve the treatment and quality of life of all pLGG patients.

## Methods

### Ethical approval
Legal guardians provided written informed consent on behalf of all pediatric patients for the use of tissues for research without compensation. All samples from McGill collaborators (CHU Sainte-Justine biobank) were collected under a protocol approved by the ethical committee of CHU Sainte-Justine. All samples from Charité (archives from the SIOP-LGG 2004 interim protocol) were collected under a protocol approved by the ethical committee of Charité Universitätsmedizin Berlin.

### Cell lines and treatments
Pilocytic astrocytoma cell lines (DKFZ-BT66, DKFZ-BT308, DKFZ-BT314, DKFZ-BT317) generated in our previous studies[45,46] were cultured in Astrocyte Basal Medium (ABM, cat. no. CC-3187, Lonza) plus supplement (Astrocyte Growth Medium BulletKit, cat.no. CC-3186, Lonza) with 1 μg/ml doxycycline (cat. no. sc-337691, Santa Cruz) to induce cell proliferation as described previously[45,46]. The BT40 cell line, kindly provided by Prof. Houghton, was grown in RPMI (cat. no. 21875091, Thermo Fisher Scientific) supplemented with 10% FCS (cat. no. S 0115, Biochrom). All cells were cultured under a 5% $CO_2$ atmosphere at 37 °C. Were indicated, DKFZ-BT317 cells transduced with the

pDIPZ reporter[48] were seeded in a 384-well plate ($1 \times 10^6$ cells per well) and treated with the corresponding MAPK inhibitor (Supplementary Data 8). Luciferase measurement was done 24 h after treatment, as described previously, and data were acquired using the OPTIMA BMG-Labtech software (v2.20R2)[48]. For combination $IC_{50}$ measurements, the $IC_{50}$ of each single drug was measured. Both drugs $IC_{50}$ were then combined, and seven dilutions (4x, 2x, 1x, 0.5x, 0.25x, 0.125x, 0.06x) of this mixture were used to draw the combination dose-response curve and measure the $IC_{50}$ of the combination.

### RNA extraction and gene expression profiling from pLGG cell lines
pLGG cell lines samples (untreated) and primary samples RNA extraction was performed using the RNeasy Mini Kit (cat. no. 74104, Qiagen) following the manufacturer's protocol. RNA was submitted to the DKFZ genomics core facility (Microarray Unit) to generate gene expression profiles with Affymetrix Gene Chip human U133 Plus 2.0. Gene expression data were then uploaded on the R2 online database (https://hgserver1.amc.nl/cgi-bin/r2/main.cgi) and MAS5.0 normalized.

### RNA sequencing on primary samples with known response to MAPKi
Fresh frozen primary material from pLGG patients with known response to MAPKi (trametinib) therapy was collected from two partner institutes (Charité, Berlin; McGill University, Montreal) (cohorts' details in Supplementary Data 6). Tumor material was surgically collected before trametinib treatment initiation. Patients' response to trametinib was assessed based on the RAPNO criteria[55].

For the samples from Charité ($n = 3$), RNA extraction was performed using the RNeasy Mini Kit (cat. no. 74104, Qiagen) following the manufacturer's protocol. cDNA synthesis and library preparation were done using the Agilent SureSelect XT HS2 RNA kit, and sequencing was done on an Illumina NovaSeq 6000 platform. For the samples from McGill University ($n = 2$), total RNA was isolated using AllPrep DNA/RNA/miRNA Universal kit (Qiagen), according to the manufacturer's instructions. RNA samples were rRNA depleted using QIAseq FastSelect (Human/Mouse/Rat 96rxns). cDNA synthesis was achieved with the NEBNext RNA First Strand Synthesis and NEBNext Ultra Directional RNA Second Strand Synthesis Modules (New England BioLabs). The remaining steps of library preparation were done using the NEBNext Ultra II DNA Library Prep Kit for Illumina (New England BioLabs), and sequencing was done on an Illumina NovaSeq 6000 platform. There was no material left over after extraction.

Fastq files were finally submitted to the RNAseqWorkflow pipeline on the OTP platform for automated reads trimming, mapping to the hg19/GRCh37 human genome, and calculation of TPM per transcript[98].

### scRNA sequencing and data analysis of primary pLGG samples
Fresh tumor material from the Münster pLGG samples ($n = 6$) was subjected to papain digestion to generate a single-cell solution. Non-vital cells were removed by 7-AAD staining (cat. no. 00-6993-50, Thermo Fisher) using a FACSAria II cell sorter (BD Biosciences), and approximately 10,000 vital cells were used as input for scRNA-seq. Single-index libraries were generated by Chromium Single Cell 3' v3.1 technology (10x Genomics) and sequenced using an Illumina NextSeq 2000 sequencing system at the Genomics Core Facility (University Hospital Münster, Germany). The samples were analyzed with the 10x Genomics CellRanger v6.0.2 pipeline and Seurat R package v4.0.5. Raw data were converted to Fastq format with the CellRanger mkfastq function and subsequently aligned against the human reference transcriptome GRCh38 v2020-A with CellRanger count and default values. Seurat objects were created for both samples based on the following filter criteria: a minimum of 3 cells, a minimum feature number of 200, and cells with <25% of mitochondrial genes. Outlier cells with a high

nCount_RNA value were classified as doublets and removed (threshold: 30,000–35,000). The filtered data were then normalized, integrated and clustered with Seurat's SCTransform routine using a resolution parameter of 0.5. Feature plots, UMAPs and dotplot visualizations were created with Seurat functions; a cluster-based cell type annotation was conducted based on the expression of characteristic marker genes per cell type.

### Other gene expression datasets

For signature generation, the Genomics of Drug Sensitivity in Cancer (GDSC) dataset was used. RMA-normalized data can be downloaded on the GDSC website (see "Data availability" section).

For signature validation in the PDX cohort, RNAseq data from the "Database For PDX Pharmacogenomic Data" (XevaDB, Novartis) was used[99]. FPKM values were converted to TPM values in order to make sample comparison possible for subsequent single-sample Gene Set Enrichment Analysis (ssGSEA), as recommended by the Broad Institute (https://software.broadinstitute.org/cancer/software/gsea/wiki/index. php/Using_RNA-seq_Datasets_with_GSEA). The type of response followed the mRECIST criteria described in the original publication. All PDXs showed a primary response, almost always followed by the emergence of treatment resistance induced by newly acquired genetic alterations. Since this is extremely rare in pLGG[100], we only considered the primary response in our study.

Processed genomic data from the Open Pediatric Brain Tumor Atlas dataset is available through the Open Pediatric Brain Tumor Atlas portal (https://github.com/AlexsLemonade/OpenPBTA-analysis).

Finally, unified TCGA sequencing data is accessible on figshare[101] (https://figshare.com/articles/dataset/Data_record_3/5330593), and mutation status was retrieved from already published work[102].

Since we aimed to identify gene signature that would be applicable for pLGG, where all driving MAPK alterations are mutually exclusive, all non-pLGG datasets (i.e., GDSC, XevaDB, TCGA) were pre-filtered to only keep samples with mutually exclusive driving MAPK alterations.

### MAPKi sensitivity gene signature generation

MAPKi sensitivity signatures were generated from the Genomics of Drug Sensitivity in Cancer (GDSC) dataset[42]. Both GDSC1 and GDSC2 drug sensitivity runs were kept in the analysis. As both GDSC1 and GDSC2 datasets were included, the same cell line could have been tested with the same inhibitor several times. For clarity in that case, we will henceforth describe these as "samples" treated with MAPKi. The data from the GDSC2 dataset was used preferentially in case of conflicting results across datasets 1 and 2, as suggested by the GDSC authors[42]. The list of cell lines comprising both sets can be found in Supplementary Data 9. The metrics described below were used to assess the signature's predictive power.

### ROC binary evaluation (Youden's J statistics & F1-score)

Measured IC50 (i.e., IC50 $z$-score) and predicted sensitivity (i.e., signature's ssGSEA score = discrimination threshold) were used to evaluate the efficiency of each signature to classify each sample into a sensitive or non-sensitive category. The IC50 $z$-score threshold was set at 0 to allow to differentiate between the samples that would respond to MAPKi (clinically equivalent to stable disease with small tumor volume reduction, to partial response or complete remission) and the samples that would not respond (clinically equivalent to stable disease with small tumor volume increase, or tumor progression). Homogenously distributed predicted sensitivity thresholds ($n = 100$) ranging from the lowest to the highest ssGSEA score and ROC curves were generated. The optimal predicted sensitivity threshold was then selected using Youden's J statistics, and the following metrics were used to assess the signature's efficacy: Youden's J statistics (to measure the difference between true positive rate (TPR) and false positive rate

(FPR)), and F1-score to assess model performance considering TPR and precision.

### Pearson's correlation coefficient

Pearson's correlation coefficient was measured to assess the linear correlation between MAPKi IC50 $z$-score and the predicted sensitivity scores in order to take into account the identification of signatures that not only differentiate between two groups (i.e., binary output) but are also capable of accurately capturing the different degrees of sensitivity of a given sample to a MAPKi (i.e., continuous output).

### Concordance analysis (concordance index)

Ideally, a MAPKi predictive signature should be related to the MAPK pathway activity level. However, MAPK pathway activity itself does not correlate well with MAPKi sensitivity[31]. We used the concordance index to control for a link between the MAPK pathway activity score (MPAS) and predictivity scores, which indicates the probability of equidirectional ranking of the signature score and the MPAS when comparing pairs of samples.

### Percentage of correct prediction (prediction accuracy)

Prediction accuracy of each signature to assign samples into four categories based on their predicted sensitivity scores (i.e., sensitive, medium-sensitive, medium-resistant, resistant) was calculated based on the measured sensitivity and the predicted sensitivity: IC50 $z$-score thresholds were assigned to define each measured sensitivity category (sensitive = IC50 $z$-score < −2; medium-sensitive = −2 < IC50 $z$-score < 0; medium-resistant = 0 < IC50 $z$-score < 2; resistant = IC50 $z$-score > 2). We subsequently identified the predicted ssGSEA sensitivity score threshold that allowed to differentiate between sensitive, medium-sensitive and medium-resistant cells in the GDSC dataset (Supplementary Fig. S4B). These thresholds (measured and predicted) were used to estimate the proportion of correctly classified samples, allowing calculation of the prediction accuracy.

### Consensus ranking

For each class of MAPKi (i.e., BRAFi Type I 1/2, BRAFi Type II, MEK1/2i, catERKi, dualERKi), the nine signatures were ranked per drug and metric. The ranks were then aggregated across drugs using average ranks to give the smallest Euclidean distance. This led to the generation of a consensus ranking for the nine signatures across drugs for each of the five metrics separately. This was followed by a consensus ranking across all five metrics, allowing to generate an overall ranking of the signatures based on their performance to predict sensitivity to all MAPKi included in the given MAPKi class.

### MAPK pathway activity score (MPAS) and MAPKi sensitivity scores (MSS)

To measure MAPK pathway activity, the previously described MPAS signature was used[36]. In order to palliate the cohort-dependent effect induced by the use of gene expression $z$-score used in the original paper, ssGSEA was performed to generate signature scores on a single-sample basis.

### Statistics and reproducibility

All statistical analyses were performed in R Studio (R Version 1.4.1103) using the following packages:

Correlation analysis and Pearson coefficient were carried out using the "stats" package (v4.2.1), and ROC analysis with the "pROC" package (v1.18.0).

Concordance indices were calculated using the "survival" package (v3.4-0).

Consensus ranking and related figures were done using the "challengeR" package (v1.0.2)[103].

Signatures overlap and Venn diagrams were generated using the "VennDiagram" package (v1.7.3).

Significance between groups was calculated using ANOVA followed by Tukey's 'Honest Significant Difference' test in the "stats" package. Multiple linear regression analysis was performed with the lm function after ensuring that the data followed a normal distribution (Shapiro-Wilk normality test).

Gene Set Enrichment Analysis (GSEA) was performed using the Broad Institute software (GSEA_4.0.3). The ssGSEA module (v10.1.0) from Gene Pattern was used to measure ssGSEA scores[104], using the parameters recommended in the documentation. Of note, ssGSEA scores were not normalized and are therefore considered as measured in arbitrary units.

The testing of the extent to which our signatures are confounded due to the microenvironmental transcriptome was done using ConfoundR (https://confoundr.qub.ac.uk/)[69].

scRNAseq data was analyzed using the "Seurat" package in R (v4.3.0). scRNAseq data from Reitman et al. was re-analyzed with the "Seurat" package using the parameters described in the original publication, and signature scores were measured using the "UCell" package (v2.0.1).

Graphical representations were done in R Studio using the "ggplot2" package (v3.4.1) for the correlation plots, waterfall plots and boxplots. The "ComplexHeatmap" package (v2.12.1) was used to generate the heatmaps.

$IC_{50}$ calculation was performed using GraphPad Prism 5 software (Version 5.01, GraphPad Software Inc., San Diego, USA), using a 4-parameter dose-response model.

For all in vitro experiments carried out in this study, experiments were carried out in three independent biological replicates as standard practice. Bulk RNAseq was performed on $n = 5$ samples, and scRNA seq was performed on $n = 6$ samples, as these were the only patient samples available at the time of the study. The rest of the analyses were carried out on publicly available datasets.

Data were excluded from the publicly available datasets used in the study based on criteria defined beforehand. These criteria were chosen to stay relevant to pediatric low-grade glioma biology (i.e., samples/cell lines with mutually exclusive genetic MAPK alteration) and to stay clinically relevant (i.e., exclusion of data generated from drugs other than MAPK inhibitors or MAPK inhibitors without a clear mode of action/already excluded from clinical trials).

All measures performed experimentally were always done in three independent biological replicates. All replicates are shown in the manuscript. All attempts at replication were successful. For in vitro experiments, cells were randomly allocated into control and experimental groups. The investigators were not blinded to allocation during experiments and outcome assessment.

Sex and/or gender were not a criterion for the study design or data interpretation. This information was not collected.

### Reporting summary

Further information on research design is available in the Nature Portfolio Reporting Summary linked to this article.

## Data availability

Gene expression data (RMA-normalized expression data) can be accessed from the GDSC website (https://www.cancerrxgene.org/gdsc1000/GDSC1000_WebResources/Home.html). Cell lines' drug response and genetic features can be downloaded at the following link (https://www.cancerrxgene.org/downloads/anova). Gene expression data and drug response from the Novartis PDX cohort can be accessed at the following address: https://www.xevadb.ca/ and in the Supplementary information from the original publication[99]. The already published gene expression profiles from pLGG cell lines[45,46,89], melanoma (GSE7127) and multiple myeloma (GSE6205) can be accessed on the R2 platform [https://hgserver1.amc.nl/cgi-bin/r2/main.cgi] (see also Supplementary Data 10 for MAS5.0 normalized data). Processed genomic data from the Open Pediatric Brain Tumor Atlas dataset is available through the Open Pediatric Brain Tumor Atlas portal [https://github.com/AlexsLemonade/OpenPBTA-analysis]. Finally, unified TCGA sequencing data is accessible on figshare[101] [https://figshare.com/articles/dataset/Data_record_3/5330593], and mutation status was retrieved from already published work[102]. Publicly available RNA sequencing datasets from melanoma primary samples with MAPKi response[105] can be accessed from the GEO platform (GSE65185). Gene expression from pediatric SEGA samples with TSC1/2 mutation[54] can be accessed from the publication and the European Genome-phenome Archive (EGAS00001003787). The RNA sequencing data from our pLGG primary samples with trametinib response (GSE222406) and scRNA sequencing data from $n = 6$ pLGG samples (GSE222850 generated in this study are both available from the GEO platform. All RAW data are available. Source data are provided with this paper.

## Code availability

The custom R script used to select the best predicting signature can be found in the Supplementary Software file.

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

## Acknowledgements

We thank Isabel Büdenbender and Lea Hofmann for their excellent technical support. We thank the Microarray Unit of the Genomics and Proteomics Core Facility, German Cancer Research Center (DKFZ), for providing excellent Expression Profiling services. We also thank the NGS Core Facility, German Cancer Research Center (DKFZ), for providing excellent RNA sequencing services. We thank the Omics IT and Data Management Core Facility (ODCF), German Cancer Research Center (DKFZ), for providing excellent support with the RNAseq data pre-processing. D.T.W.J., O.W. and T.M. were supported by The Everest Centre for Low-grade Paediatric Brain Tumours (GN-000707, The Brain Tumour Charity, UK). T.M., O.W., D.C. and T.B. were supported by DKTK JF Upgrade Next Gen LOGGIC (B310-JF-LOGGIC-MDE). T.B. was supported by the DFG through a Heisenberg professorship (BR 3662/5–1) and SFB-1479 Oncoescape–Project ID: 441891347 (P14). D.U. received a scholarship from the Friedrich Ebert Foundation ("Friedrich-Ebert-Stiftung").

## Author contributions

R.S. and T.M. conceived the ideas and designed the study. R.S., D.K., C.H. and N.W. performed the experiments and data analysis. T.H. provided input on biostatics and MSS evaluation strategy. T.K.A., C.W., D.M., J.V. and K.K. performed scRNA seq experiments and analysis. A.B., M.H., C.T., D.C., U.W.T., P.H.D., M.S., S.Horn, N.A.H., A.K., N.J., M.U.S, A.F.A., A.N.S.vM. and E.H. provided fresh frozen material from pLGG patients and clinical annotations. A.F.A., N.J., F.Sahm and S.Hamelmann performed RNAseq experiments. F.Selt, D.U., J.E., T.B., C.M.vT., S.M.P., O.W. and D.T.W.J. provided critical input on the project. All authors contributed to writing and reviewing the manuscript.

## Funding

## Competing interests

O.W. and T.M. were supported by research grants from Biomed Valley Discoveries, Inc., and Day One Biopharmaceuticals. The remaining authors declare no competing interests.

## Additional information

Romain Sigaud [1,2,3] ✉, Thomas K. Albert [4], Caroline Hess [1,2,3,5], Thomas Hielscher[6], Nadine Winkler [1,2,3,7], Daniela Kocher[1,2,3,7], Carolin Walter[8], Daniel Münter [4], Florian Selt[1,2,3,9], Diren Usta[1,2,3,9], Jonas Ecker[1,2,3,9], Angela Brentrup[10], Martin Hasselblatt[11], Christian Thomas [11], Julian Varghese[8], David Capper[12,13], Ulrich W. Thomale[14], Pablo Hernáiz Driever [15], Michèle Simon [15], Svea Horn[15], Nina Annika Herz[15], Arend Koch[13], Felix Sahm [16,17], Stefan Hamelmann[16,17], Augusto Faria-Andrade [18], Nada Jabado [18,19,20], Martin U. Schuhmann [21], Antoinette Y. N. Schouten-van Meeteren[22], Eelco Hoving [22], Tilman Brummer [23], Cornelis M. van Tilburg [1,2,3,9], Stefan M. Pfister[1,3,9,24], Olaf Witt[1,2,3,9], David T. W. Jones [1,25], Kornelius Kerl [4] & Till Milde [1,2,3,9] ✉

[1]Hopp Children's Cancer Center Heidelberg (KiTZ), Heidelberg, Germany. [2]Clinical Cooperation Unit Pediatric Oncology, German Cancer Research Center (DKFZ) and German Consortium for Translational Cancer Research (DKTK), Heidelberg, Germany. [3]National Center for Tumor Diseases (NCT), Heidelberg, Germany. [4]Department of Pediatric Hematology and Oncology, University Children's Hospital Münster, Münster, Germany. [5]Faculty of Biochemistry, Heidelberg University, Heidelberg, Germany. [6]Division of Biostatistics, German Cancer Research Center (DKFZ) and German Consortium for Translational Cancer Research (DKTK), Heidelberg, Germany. [7]Faculty of Biosciences, Heidelberg University, Heidelberg, Germany. [8]Institute of Medical Informatics, University of Münster, Münster, Germany. [9]Department of Pediatric Hematology and Oncology, Heidelberg University Hospital, Heidelberg, Germany. [10]Neurosurgery Dept., University Hospital Münster, Münster, Germany. [11]Institute of Neuropathology, University Hospital Münster,

Münster, Germany. [12]Berlin Institute of Health, Anna-Louisa-Karsch-Straße 2, 10178 Berlin, Germany. [13]Charité - Universitätsmedizin Berlin, corporate member of Freie Universität Berlin and Humboldt-Universität zu Berlin, Department of Neuropathology, Berlin, Germany. [14]Charité - Universitätsmedizin Berlin, Corporate member of Freie Universität Berlin and Humboldt-Universität zu Berlin, Department of Pediatric Neurosurgery, Berlin, Germany. [15]Charité - Universitätsmedizin Berlin, Corporate member of Freie Universität Berlin and Humboldt-Universität zu Berlin, German HIT-LOGGIC-Registry for pLGG in children and adolescents, Department of Pediatric Oncology and Hematology, Berlin, Germany. [16]Department of Neuropathology, Heidelberg University Hospital, Heidelberg, Germany. [17]Clinical Cooperation Unit Neuropathology, German Consortium for Translational Cancer Research (DKTK), German Cancer Research Center (DKFZ), Heidelberg, Germany. [18]Department of Human Genetics, McGill University, Montreal, QC H3A 0C7, Canada. [19]Division of Experimental Medicine, Department of Medicine, McGill University, Montreal, QC H4A 3J1, Canada. [20]Department of Pediatrics, McGill University, and The Research Institute of the McGill University Health Centre, Montreal, QC H4A 3J1, Canada. [21]Section of Pediatric Neurosurgery, Department of Neurosurgery, University Hospital Tübingen, Tübingen, Germany. [22]Princess Màxima Center for Pediatric Oncology, Utrecht, The Netherlands. [23]Institute of Molecular Medicine and Cell Research (IMMZ), Faculty of Medicine, University of Freiburg, Freiburg, Germany, Centre for Biological Signaling Studies BIOSS, University of Freiburg and German Consortium for Translational Cancer Research (DKTK), Freiburg, Germany, German Cancer Research Center (DKFZ), Heidelberg, Germany. [24]Division of Pediatric Neurooncology, German Cancer Consortium (DKTK), German Cancer Research Center (DKFZ), Heidelberg, Germany. [25]Division of Pediatric Glioma Research, German Cancer Research Center (DKFZ), Heidelberg, Germany. ✉e-mail: r.sigaud@kitz-heidelberg.de; t.milde@kitz-heidelberg.de

