## [Peer Review File · Nature Communications]

REVIEWER COMMENTS

Reviewer #1, expertise in paediatric brain cancer genomics (Remarks to the Author):

In the manuscript entitled “ MAPK inhibitor sensitivity scores (MSS) predict sensitivity to MAPK inhibitors and uncover immune infiltration driving sensitivity in pediatric low-grade gliomas” the authors developed elegant predictive tools for the stratification of pLGG patients (MSSs), however, these tools are still too preliminary and have not been fully validated. The paper is well organized with consistent bioinformatic analyses, however I do not see enough data to support their claims. As written by the authors in line 314 “Prospective validation of the MSSs in clinical trials including larger patient cohorts treated with the respective MAPKi, is needed.” Since the paper is based on new tools for patient stratification, the analysis of more MAPKi treated pLGG patients is mandatory (only a few are present in the manuscript, i.e. Figure 4D, the full range of the Y-axis should be provided). In my opinion, the strength of the paper should be the validation of the MSS, which is currently missing.

Furthermore, to further add novelty to the paper it would be interesting to see the effect of the combination of several drugs in in-vitro models/patients with pLGG.

Reviewer #2, expertise in scRNA-seq, bioinformatics and drug response (Remarks to the Author):

Major issues

Sigaud et al. propose a gene expression signature to predict response to MAPK inhibitors (MAPKi) in pediatric low-grade gliomas (PLGGs). The authors validate their signature obtained from experiments on GDSC cell lines in PDX models and on the public TGCA and OPBTA datasets. In general, the work is technically well done and there is a valuable effort since the authors provide a gene expression signature with the potential to define candidates for treatment with MAPKi in pLGG patients.

Although interesting and valuable, the work is very specific as it focuses on a unique expression signature in response to one particular type of treatment (MAPKi) for PLGGs tested in silico on external datasets. This study uses large pharmacogenomics screening projects such as GDSC with hundreds of drugs tested in cancer cell lines so the author's approach should be extended to other Drug-associated gene expression signatures. On the other hand, the authors perform the analysis of gene expression signatures following usual approaches such as GSEA, ssGSEA and ConfoundR. It is not clear to me what the contribution is from the methodological point of view. Since the article is entirely focused on the transcriptomics data analysis, this is a major point and should be clarified in the manuscript.

Minor issues

The manuscript is easy to understand, although sometimes there are many acronyms and the reading becomes a little uncomfortable.

Reviewer #3, expertise in MAPK signalling, kinase inhibitors and GBM therapy (Remarks to the Author):

This is an interesting and relevant study with clear clinical relevance.

The investigators have effectively developed a robust scoring system for multiple MAPK inhibitor classes, and clearly demonstrated an improved predictive ability compared to the prior scoring system. There are a few areas which could be clarified to improve the readability of this manuscript and a few grammatical and spelling errors, but the experimental plan and conclusions are good.

Section comments:

The abstract is well written and covers the key points of the manuscript, whilst the introduction appropriately describes the prior knowledge in this field and explains the importance of the research question.

The results section is generally clear, however there are a few abbreviations which aren't explained, for example catERK, PA and PXA, which reduces the readability.

Additionally, for the section describing the results in Figure 2 it is unclear that the significant result from encorafenib is a true result given the extremely small proportion of patients who achieved a partial or complete response increases the possibility that this result is due to chance, especially as the pattern of MSS is in keeping with the other inhibitors.

The significance levels are not clearly documented in this grouped section of the figure whereas the significance level is clearly documented in the ungrouped section. The section describing Figure 8B

separates the groups into Cluster High and Cluster Intermediate/Low, however there seems to be a more significant difference separating Cluster Low on its own with a lesser split between Cluster High and Intermediate. This difference would further highlight the authors point and should be included to highlight the negative association between MSS and neuronal features.

I would also encourage consistency of ordering throughout the figures, including the order of MAPK inhibitor sensitivity scores in Figure 1H and Supplemental Figures 3, 5A, compared to the remaining Figures, the order of MAPK inhibitors in Figure 2 compared to Supplemental Figure 7, and the positioning of the normal tissue results in Supplemental Figure 12.

The authors should also review the labelling of Supplemental Figure 11 and consider if alphabetical is necessarily the clearest way to organise Supplemental Figure 12F.

The discussion is generally well written, although it is unclear if the authors have already performed the validation experiments described in lines 536-7 and this should be clarified.

Reviewer #4, expertise in TIME in GBM (Remarks to the Author):

Sigaud et al provide very nice work to come to four gene signatures, that they term MAPKi Sensitivity Scores (MSS) and that are MAPKi class-specific. These MSSs appear to correlate robustly with MAPKi responses in vitro and in vivo (PDX models). In pLGG patients, a high MSS score correlated with a higher immune infiltrate in the tumors and related to the higher expression of MSS in tumor-infiltrating microglia. Importantly, the MSS score could be predictive of MAPKi therapy response and could be a future tool to stratify patients. Some remarks remain:

- An aspect that needs clarification is the fact that the four MSSs were established in vitro and based on cell line studies. However, ultimately, and somewhat surprisingly in the flow of the story, it turns out that tumor-infiltrating microglia may be important mediators of the MAPKi responsiveness in CNS tumors. Since microglia were not present in the initial screening for signatures, it seems to me that important signature genes may actually be missing and that these scores could be improved if signatures would have been established based on in vivo (eg PDX) data.

- Related to this, it would be nice to establish the relative importance of MSSs within microglia versus within the cancer cells in relation to MAPKi responsiveness. Can the authors find patients in which MSS is high in cancer cells, but not in microglia/macrophages, or vice versa? Would this lead to a different outcome for the MAPKi treatment? The essential question here is obviously: which cells are actually the crucial ones to mediate MAPKi responses?

- Fig 6A. CX3CR1 is not bad, but TMEM119 is a somewhat "cleaner" marker to identify microglia. No cDC1 were detected (XCR1 expression)? Did you consider cells that co-express CD3G and PRF1, as these are likely NKT cells?

- Statements in lines 381-385 require some nuance. MSSs are clearly also high in monocyte-derived TAMs, albeit somewhat less high than in tumor-associated microglia. Hence, MSS is not "specifically" elevated in tumor-associated microglia, and it can not be excluded that monocyte-derived TAMs may contribute to MAPKi efficiency, especially in non-CNS tumors. It is also true that the correlation between MSSs and ESTIMATE is highest in CNS tumors, but there is never a significant difference with non-CNS tumors. In conclusion, statements need to be tempered and other possibilities need to be acknowledged.

Response to reviewers' comments on NCOMMS-22-52491-T

Reviewer #1, expertise in paediatric brain cancer genomics

In the manuscript entitled “ MAPK inhibitor sensitivity scores (MSS) predict sensitivity to MAPK inhibitors and uncover immune infiltration driving sensitivity in pediatric low-grade gliomas” the authors developed elegant predictive tools for the stratification of pLGG patients (MSSs), however, these tools are still too preliminary and have not been fully validated. The paper is well organized with consistent bioinformatic analyses, however I do not see enough data to support their claims.

We would like to thank Reviewer 1 for their careful review of our manuscript and important comments that have allowed to significantly improve the manuscript's relevance and clarity. We will address the reviewer's comments below.

Comment 1:

As written by the authors in line 314 “Prospective validation of the MSSs in clinical trials including larger patient cohorts treated with the respective MAPKi, is needed.” Since the paper is based on new tools for patient stratification, the analysis of more MAPKi treated pLGG patients is mandatory (only a few are present in the manuscript, i.e. Figure 4D, the full range of the Y-axis should be provided). In my opinion, the strength of the paper should be the validation of the MSS, which is currently missing.

We fully agree with Reviewer 1 about the fact that validation of the MSS in a larger pLGG cohort is mandatory before any prospective application in clinical trials. As noted by the reviewer, we have presented data on a total of 5 cases, correlating MSS (RNA expression) and MAPKi response (trametinib) in Fig. 4D, 2 of which were obtained through the SIOP LOGGIC registry and LOGGIC Core (Berlin), the European-wide pLGG registry and bio-molecular databank, and 3 of which were from patients treated at McGill, Montreal, Canada. Considering that a) treatment with MAPKis of patients with pLGGs has started only within the last 10 years (first CT investigating vemurafenib in BRAF^{V600E} driven pLGG in 2014: NCT01748149), mostly within clinical trials, and that b) collection of fresh frozen tissue (as a pre-requisite for RNA sequencing) has only been done in an increasing manner in the past years, it is indeed not self-evident that a number of cases is available today for this analysis combining a) response after MAPKi treatment, and b) RNA seq data.

To identify further suitable datasets for analysis, i.e. corresponding RNA expression and MAPKi response data, we have inquired with the largest clinical trials to date investigating MAPKis in pLGGs (specifically for the matter of this revision: Jason R. Fangusaro - PBTC-029 trial – MEKi selumetinib; and Samuel C. Blackman – FireFly 1 trial – Type II RAFi tovorafenib, had been inquired by us prior to the submission of the manuscript). All PIs confirmed that no data on RNA expression (regardless of technique) from patients on the trials was obtained or available, neither was material available suitable for post-hoc analysis. Thus, no data on RNA expression to be correlated with response (to MAPKi) data was available from any of these clinical trials, which are the largest and most recent ones on MAPKis in pLGGs. As outlined above, this is unfortunately not surprising, as the clinical trials investigating MAPKis in pLGGs have only been initiated recently, and (as confirmed by the PIs of the respective trials) without collection of biological material suitable for RNA analyses for research purposes. This data will hopefully be available in the future.

However, we would like to clarify that we are currently not suggesting to use the MSS as stratification tools just yet, but rather describe the MSS as potential tools that will need further validation (e.g. to identify thresholds applicable in the clinic) before any clinical use. This will be investigated in upcoming clinical trials, as discussed in the "Discussion" section line 314. In order to avoid misunderstanding of potentially misleading wording, we have now changed the text to "this clearly highlights the potential for such scores to be further explored and validated in future clinical trials" line 534-535, and "Larger gene expression datasets [...]are now urgently needed to identify and validate sensitivity score thresholds to assess the applicability of our MSS as stratification tools for clinicians." line 622-624.

Nonetheless, to address the reviewer's comment on the lack of validating data, we have now added further data to strengthen the clinical relevance of our MSS supporting further investigation, as well as validation in upcoming pLGG clinical trials. We used a publicly available RNA sequencing dataset from melanoma patients with mutually exclusive MAPK alterations (to stay true to the single MAPK alteration genotype in pLGG biology) and with known response to treatment with the BRAFi Type I½ vemurafenib. We could show that the responders had a median BRAFi Type I½ MSS higher than the non-responders, with a difference almost reaching significance despite the small number of samples (Fig. 1). This data adds further value to our MSS and their relevance to be further evaluated for clinical use.

Fig. 1 Independent validation of MSSs in a melanoma cohort with RNAseq data at baseline coupled with vemurafenib response

We have added this new data to our manuscript as supplementary figure S14, and added it to the result section "To address the current lack of clinical response data of pLGG patients to MAPKi paired with RNAseq data from baseline, we analyzed a publicly available RNA sequencing dataset from melanoma patients, with mutually exclusive MAPK alterations and which were treated with the BRAFi Type I½ vemurafenib. Notably, the data showed that patients who responded well to the treatment had a higher median BRAFi Type I½ MSS than those who did not respond, and this difference was almost significant despite the small sample size (Suppl. Fig. S14). This finding supports the notion that the MSS might be useful in predicting response to treatment and should be further evaluated and validated for clinical use." (line 326-333) and discussion "Furthermore, we were able to demonstrate a similar pattern in an independent melanoma dataset with response to the BRAFi Type I½ vemurafenib." (line 532-533).

Regarding the reviewer's point "[...] i.e. Figure 4D, the full range of the Y-axis should be provided".

Concerning the depiction of the partial y-axis range shown in Fig. 4D, this indeed touches an important point. As shown in e.g. Fig. 4A, the range of MSS (y-axes) is within a relatively high absolute number (MSS in arbitrary units), and scores are high on an absolute scale, even for tumor samples with relatively low score and control tissues like EWS. This underscores that the MSS is a relative score, and future studies will need to establish the technical aspects of the MSS such as range, relative and absolute comparability, cut-off values and negative and positive predictive values, as discussed in our discussion line 552-556 "Another challenge is the comparison of scores among themselves, as it is at the moment not possible to compare ssGSEA scores (i.e. MSSs) with each other, which would allow to estimate whether one MSS is higher or lower than another. Some approaches could be envisioned to circumvent these pitfalls, such as a resampling procedure to generate null distributions for each of the MSSs, which will need further validation in prospective studies.", and line 621-624 "Larger gene expression datasets derived from patients that received a MAPKi treatment, coupled with their respective response to the treatment (such as in the upcoming LOGGIC Trial), are now urgently needed to identify and validate sensitivity score thresholds to assess the applicability of our MSS as stratification tools for clinicians". Of note, the output ssGSEA scores used to depict our MSS are by default not normalized, and are therefore considered as measured in arbitrary units. Since these scores are analyzed in relation to each other, and not in relation to a reference (i.e. 0), we have kept the original depiction of absolute scores (also for the purpose of data clarity and transparency) and did not show the full Y-axis range in Fig. 4D to avoid skewing the data. This is now clarified in the method section: "Of note, ssGSEA scores were not normalized, and are therefore considered as measured in arbitrary units." (line 772-773).

Comment 2:

Furthermore, to further add novelty to the paper it would be interesting to see the effect of the combination of several drugs in in-vitro models/patients with pLGG.

We agree with Reviewer 1 that exploration of the applicability of MSS to combination treatments is of high relevance for clinical application, where patients are likely to receive a combination treatment of several MAPKi.

In order to address this comment, we first treated two of our pLGG models (DKFZ-BT314 with a BRAF^{V600E} mutation, DKFZ-BT317 with a KIAA1549:BRAF fusion) with the first FDA approved systemic therapy for first-line treatment of pediatric patients with pLGG with a BRAF V600E mutation, i.e. trametinib + dabrafenib. Since the BRAFi Type I 1/2 dabrafenib is likely to induce MAPK pathway paradoxical activation in the KIAA1549:BRAF fusion cells (i.e. BT317), we also tested a combination of the MEKi trametinib with the clinically investigated BRAFi Type II tovorafenib (DAY101). For each combination, cells were treated with 7 dilutions (4x, 2x, 1x, 0.5x, 0.25x, 0.125x, 0.06x) of both their respective IC₅₀ to assess the combination dose-response and measure the IC₅₀ of the combination, as a surrogate for combination sensitivity. Since the BT314 model was not included in the initially submitted manuscript, we measured its corresponding IC₅₀s for trametinib, dabrafenib and tovorafenib. Of note, for data completion purposes, we extended our previous comprehensive dataset of MAPKi dose response in pLGG models, now including the same dose response data for all MAPKis also in DKFZ-BT314, and included the data in Suppl. Fig. S10, and Fig. 3B, with similar and in summary unchanged conclusions.

The data obtained show an inverse correlation between the measured combination $\log_{10}(\text{IC}_{50})$ and the predicted sensitivity to both inhibitor classes, MEK1/2i and BRAFi Type I½ (i.e. increased MAPKi sensitivity correlates with higher MSS) (Fig. 2).

Fig. 2 Correlation of BRAFi Type I½ + MEKi combination response and MSSs in pLGG models in vitro

In addition, we investigated the MSS in two other pLGG models (BT40 with a BRAF^{V600E} mutation, DKFZ-BT66 with a KIAA1549:BRAF fusion) treated with a combination a catalytic ERKi (ulixertinib) and several MEKi (trametinib, selumetinib and binimetinib) from a study that we recently published.¹ The same pattern was observed, with the combination $\log_{10}(\text{IC}_{50})$ being highest for all three combinations in the model with the lowest MSS, and a decreased $\log_{10} \text{IC}_{50}$ with increasing MSS (Fig. 3).

Fig. 3 Correlation of MEKi + catERKi combination response and MSSs in pLGG models *in vitro*

As discussed above (see response to comment 1), there is unfortunately no RNA data from pLGG samples from patients treated with a combination of MAPKi that we could have used to correlate with response data. We have inquired with the PI (Marc Russo) of the Tadpole-G clinical trial investigating MAPKi combinations (BRAFi Type I 1/2 dabrafenib + MEKi trametinib) in pLGGs, who again confirmed that no data on RNA expression (regardless of technique) from patients on the trial was obtained or available, neither was material available suitable for post-hoc analysis (personal communication). In absence of clinical patient data, we analyzed the MSS in an *in vivo* setting as an alternative approach. Using data from the XevaDB of *in vivo* PDX treated with a combination of the MEKi binimetinib and the BRAFi Type I½ encorafenib, we can show that the MSS is positively associated with response to the combination treatment, i.e. the higher the MSS, the better the PDX respond to treatment (Fig. 4).

Fig. 4 Correlation between MAPKi sensitivity scores and MEKi+BRAF Type I½ combination response in vivo

Of note, it appears that the best responders are the PDX samples with high MSSs for both drugs, as opposed to high MSS for only one drug, which may suggest that patients with highest MSS for two drugs might be more likely to benefit from a combination therapy of those two drugs than those with high MSS for only one of the drugs. This observation will of course need further validation in upcoming clinical trials.

These data have now been included in the manuscript as Supplementary figure S16, and in the text line 342-361: “We also investigated whether the MSS could be useful in predicting response to combination treatments, as patients are often treated with a combination of multiple MAPKis. To do so, we first treated two pLGG models (DKFZ-BT314: BRAF^{V600E} mutation; DKFZ-BT317: *KIAA1549:BRAF* fusion) with the first FDA approved systemic therapy for first-line treatment of pediatric patients with pLGG with a BRAF^{V600E} mutation, namely trametinib and dabrafenib. Since dabrafenib is not recommended for the treatment of *KIAA1549:BRAF* fusion pLGG, we also tested a combination of the MEKi trametinib with the currently clinically investigated BRAFi Type II tovorafenib (DAY101). For both of these BRAFi+MEKi combinations, we could show an inverse correlation between the measured combination log₁₀(IC₅₀) and the predicted sensitivity to both combined drugs, i.e. increased MAPKi sensitivity correlated with increased MSS (Suppl. Fig. S16A). Other combinations were assessed in recently published data on combinations of the first-in-class catalytic ERKi (ulixertinib) and several MEKi (trametinib, selumetinib, and binimetinib) in several pLGG models (BRAF^{V600E} mutation and *KIAA1549:BRAF* fusion).¹ Using these data, we could show the same correlation between increased sensitivity to the combination treatment and increased MSS (Suppl. Fig. S16B). Finally, using data from the XevaDB of in vivo PDX treated with a combination of the MEKi binimetinib and the BRAFi Type I½ encorafenib, we showed that the MSS for each drug was positively associated with the response to the combination treatment, i.e. the higher the MSS, the better the PDX responded (Suppl. Fig. S16C), with the best responses in samples with high MSSs for both drugs. In summary, these findings suggest that the MSS might be useful in predicting response to combination treatments as well, and may be a valuable tool for clinical combination therapies.” and in the discussion line 535-540 “It is worth noting that in the context of combination therapy, our

MSS was also positively associated with the response to several types of MAPKi in vitro and in PDXs in vivo. It appeared the best responders were the samples with highest MSS for both drugs (as opposed to high MSS for only one drug), which may suggest that patients with high MSS for two drugs might be more likely to benefit from a combination therapy. This observation will certainly need further validation in upcoming clinical trials.”

Reviewer #2, expertise in scRNA-seq, bioinformatics and drug response

Major issues

Sigaud et al. propose a gene expression signature to predict response to MAPK inhibitors (MAPKi) in pediatric low-grade gliomas (PLGGs). The authors validate their signature obtained from experiments on GDSC cell lines in PDX models and on the public TGCA and OPBTA datasets. In general, the work is technically well done and there is a valuable effort since the authors provide a gene expression signature with the potential to define candidates for treatment with MAPKi in pLGG patients.

We would like to thank Reviewer 2 for their encouraging appreciation of the manuscript, and for the valuable comments that we will address below.

Comment 3:

Although interesting and valuable, the work is very specific as it focuses on a unique expression signature in response to one particular type of treatment (MAPKi) for PLGGs tested in silico on external datasets. This study uses large pharmacogenomics screening projects such as GDSC with hundreds of drugs tested in cancer cell lines so the author's approach should be extended to other Drug-associated gene expression signatures.

As highlighted by Reviewer 2 here, our work was indeed focused on predicting sensitivity towards MAPKi in pLGG only, since these drugs are overwhelmingly the main targeted therapy agents currently investigated in clinical trials in pLGG.

However, we fully appreciate Reviewer 2's comment, that it would be appropriate to generate additional drug-associated gene expression signatures, as this would validate the concept of our approach on a technical level, and potentially generate new signatures worth further investigation in follow up studies focusing on other targeted therapies. We therefore generated three new exemplary signatures for n=3 drugs with a clinically well-described drug-target relation, which might be of potential interest for the treatment of pLGG in future studies, i.e. NTRKi, FGFRi, and mTORi. We used the same pipeline that we used for our MSS signatures in order to generate these new drug-associated signatures.

As described previously, the GDSC dataset was again randomly split in a first step, then grouped based on their sensitivity (IC50 z-score) in a second step, and a GSEA was performed to identify the genes significantly associated with the respective drug sensitive cell lines, belonging to a gene signature related to the drug on interest (to avoid selecting background genes specific to the tissue the cells were derived from). These genes were used to measure the putative drug sensitivity score via ssGSEA. Drug prediction efficacy metrics (from correlation and ROC analyzes) were measured in order to rank the signatures based on their ability to predict drug sensitivity in both Testing and Validation sets. The heatmaps summarizing the rankings of the signatures identified are shown below (Fig. 5).

Fig. 5 Validation of the gene signature pipeline by generating gene expression-derived signature for other pLGG relevant drug classes: NTRKi, FGFRi, mTORi.

The same signatures topped the rank in both training and validation sets, showing the validity of this approach to generate drug-associated gene signatures also in other settings than MAPKi. The mTORi signature score was the highest in a SEGA primary sample dataset with TSC1/2 mutations, and was higher in an mTORi responding patient compared to a patient who did not respond to mTORi therapy. The NTRKi signature score was the highest in primary pLGG samples with NTRK fusions from the Open Pediatric Brain Tumor Atlas (OPBTA). The FGFRi signature was the highest in the subset of pLGG with FGFR alterations and normal tissue (only n=2 samples), the latter potentially because of high expression of the FGFR2 in at least one of these two samples (Fig. 6).

Fig. 6 Sensitivity score to the corresponding drug signatures evaluated by ssGSEA in a SEGA dataset and the OPBTA dataset.

The data has now been added as Supplementary figure S7, and further described and discussed in the text “In order to illustrate the relevance of our approach, we generated novel signatures for three drug classes which might be of potential interest for the treatment of pLGG in future studies, i.e. NTRKi, FGFRi, and mTORi. Since the cell lines comprising the dataset were not harboring relevant genetic alterations (NTRK fusions, FGFR fusions or mutations, TSC1/2 mutations), the GDSC dataset was randomly split regardless of the underlying genetic background. Sensitivity data from specific NTRKi (GW441756, AZD1332), FGFRi (AZD4547, PD173074), and mTORi (AZD2014, AZD8055, Temsirolimus, Rapamycin) were used. GSEA was performed using pathway specific gene signatures (“REACTOME_SIGNALING_BY_NTRKS”, “REACTOME_DOWNSTREAM_SIGNALING_OF_ACTIVATED_FGFR”, and “HALLMARK_PI3K_AKT_MTOR_SIGNALING”, respectively). The same metrics as for the MSS were used to rank the signature, except for the concordance with MAPK pathway activity. Gene signatures topping the ranking in both Discovery and Testing sets were identified, potentially identifying signatures predicting sensitivity to mTORi (Suppl. Fig. S7A), NTRKi (Suppl. Fig. S7B), and FGFRi (Suppl. Fig. S7C). Interestingly, the mTORi signature score was the highest in a set of pediatric subependymal giant cell astrocytomas (SEGA)

with TSC1/2 mutation compared to normal brain tissue from Bongaarts et al. (Suppl. Fig. S7D).² The NTRKi signature score was the highest in primary pLGG samples with NTRK fusions from the OPBTA (Suppl. Fig. S7E). Finally, the FGFRi signature was the highest in the subset of pLGG with FGFR alterations compared to pLGGs with other alterations, and two normal tissue samples (Suppl. Fig. S7F), the latter potentially because of high expression of FGFR2 (comprised in the gene signature) in these samples (Suppl. Fig. S7G). Taken together, these data validate our approach of sensitivity gene signature identification, and identify new gene signatures (Suppl. Table S4) potentially associated with response to NTRKi, FGFRi and mTORi, which could be further explore in subsequent studies.” line 212-234 and in the discussion “Importantly, we were able to apply this approach to other pLGG relevant drugs (e.g. NTRKi, FGFRi, and mTORi), generating new signatures that could be further explored in future studies.” line 506-507.

Comment 4:

On the other hand, the authors perform the analysis of gene expression signatures following usual approaches such as GSEA, ssGSEA and ConfoundR. It is not clear to me what the contribution is from the methodological point of view. Since the article is entirely focused on the transcriptomics data analysis, this is a major point and should be clarified in the manuscript.

We thank the Reviewer 2 for this valuable comment which will add further clarity to the manuscript. Using gene signatures to predict drug sensitivity is indeed not novel, as several studies have already shown the relevance and efficacy of such signatures.³⁻⁵ We have used the pre-existing methods mentioned in the comment to the following purposes: 1) GSEA: to identify MAPK-related genes enriched in MAPKi sensitive cell lines to generate our MSS gene signatures, 2) ssGSEA: to measure the MSS in several sample types (cell lines, primary material), 3) ConfoundR: to test whether our signatures were confounded by microenvironment/stromal related genes.

We agree that the methods used in our approaches have been described before, adding the advantage of previously validated methods to test and measure the performance of our newly described MSS.

We believe that this methodical validation of the gene signatures is highly important for the following reasons: it is well established that MAPK pathway inhibition is highly dependent on the tumor type, as for instance melanoma will rapidly develop BRAFi resistance contrary to pLGG,⁶ despite similar driving alterations (here: BRAF V600E). In addition, tumor response to MAPKi can vary not only across different types of genetic MAPK alterations, but also within the same (identical) genetic MAPKi alteration.⁷⁻⁹ Therefore, while using the most appropriate tool to find a gene signature is crucial, it is even more important to identify and use the most specific gene signature. In our case, we wanted to generate the most accurate and unbiased signature that captures MAPKi sensitivity in pLGG tumors. This is why we believe that the real novelty of our work lies in the actual unique MAPKi sensitivity signatures, rather than in the individual tools we've used these signatures with. Importantly, we still recognize the importance of the combination of the methodical tools, as laid out above. We have now clarified this in the discussion line 507-516 “This highlights the relevance of such biology-driven approach to identify new drug-sensitivity signatures from large pharmaco-genomics databases. While our approach didn't use nor generate new tools to measure sensitivity score per se, the combination of the methods used allowed us to generate specific and unbiased gene signatures, that may capture MAPKi sensitivity in pLGG

tumors.”

Comment 5:

Minor issues

The manuscript is easy to understand, although sometimes there are many acronyms and the reading becomes a little uncomfortable.

We also thank the Reviewer 2 for this important feedback. We have now spelled out the acronyms more frequently in the text (e.g. line 252, 278, 313-315, 438), which we hope will make the reading more pleasant.

Reviewer #3, expertise in MAPK signalling, kinase inhibitors and GBM therapy

This is an interesting and relevant study with clear clinical relevance.

Comment 6:

The investigators have effectively developed a robust scoring system for multiple MAPK inhibitor classes, and clearly demonstrated an improved predictive ability compared to the prior scoring system. There are a few areas which could be clarified to improve the readability of this manuscript and a few grammatical and spelling errors, but the experimental plan and conclusions are good.

We thank Reviewer 3 for their positive and encouraging feedback on our work, pointing out a few grammatical and spelling errors. We went through the manuscript and corrected them appropriately. We hope the current version has improved.

Comment 7:

Section comments:

The abstract is well written and covers the key points of the manuscript, whilst the introduction appropriately describes the prior knowledge in this field and explains the importance of the research question.

The results section is generally clear, however there are a few abbreviations which aren't explained, for example catERK, PA and PXA, which reduces the readability.

We thank the reviewer for this useful comment. We have now added the missing explanation to the missing abbreviations. We hope this will increase clarity of the manuscript.

Comment 8:

Additionally, for the section describing the results in Figure 2 it is unclear that the significant result from encorafenib is a true result given the extremely small proportion of patients who achieved a partial or complete response increases the possibility that this result is due to chance, especially as the pattern of MSS is in keeping with the other inhibitors. The significance levels are not clearly documented in this grouped section of the figure whereas the significance level is clearly documented in the ungrouped section.

We appreciate the reviewer's comment about the lack of clarity of the statistical approach used in the lower right panel of the sub-figures in Fig. 2. We have now added the test statistic used to measure the pval (ANOVA followed by the Tukey's 'Honest Significant Difference' test) in the figure legend.

Comment 9:

The section describing Figure 8B separates the groups into Cluster High and Cluster Intermediate/Low, however there seems to be a more significant difference separating Cluster Low on its own with a lesser split between Cluster High and Intermediate. This difference would further highlight the authors point and should be included to highlight the negative association between MSS and neuronal features.

We appreciate the reviewer's suggestion, and fully agree with it. We have now adapted the results section as follows: "The samples from the MSS low clusters had a neuronal score significantly higher than the samples with a high/intermediate MSS" line 459-460.

Comment 10:

I would also encourage consistency of ordering throughout the figures, including the order of MAPK inhibitor sensitivity scores in Figure 1H and Supplemental Figures 3, 5A, compared to the remaining Figures, the order of MAPK inhibitors in Figure 2 compared to Supplemental Figure 7, and the positioning of the normal tissue results in Supplemental Figure 12.

We thank reviewer 3 for his comment that improved consistency across the figures. We have now re-arranged the MSS in Fig. 1H, Suppl. Fig. S3 and S5A to have the same order as the rest of the figures (i.e. BRAFi Type I½, BRAFi Type II, MEKi, dualERKi, catERKi). We have re-ordered the MEKi in Suppl. Fig. S7 (now Suppl. Fig. S8) to have the same order than Fig. 2. We have also repositioned the normal tissue in Suppl. Fig. S12 (now Suppl. Fig. S15) so that the order remains consistent.

Comment 11:

The authors should also review the labelling of Supplemental Figure 11 and consider if alphabetical is necessarily the clearest way to organize Supplemental Figure 12F.

We thank reviewer 3 for his rigorous examination of the figures and the helpful comments. We have reviewed the labelling of Suppl. Fig. S11 (now Suppl. Fig. S13) and have re-arrange the legend in alphabetical order. We have also followed the reviewer's advice and re-organized Suppl. Fig. S12F (now Suppl. Fig. S15F) by increased average MSS.

Comment 12:

The discussion is generally well written, although it is unclear if the authors have already performed the validation experiments described in lines 536-7 and this should be clarified.

After review of this paragraph, it appeared that the sentence "Validation of such observation on our independent pLGG scRNA-seq dataset allowed to rule-out potential confounding factor." was misplaced, and probably resulted from a copy/paste error. The sentence has now been removed. We thank reviewer 3 for pointing this out.

Reviewer #4, expertise in TiME in GBM

Sigaud et al provide very nice work to come to four gene signatures, that they term MAPKi Sensitivity Scores (MSS) and that are MAPKi class-specific. These MSSs appear to correlate robustly with MAPKi responses in vitro and in vivo (PDX models). In pLGG patients, a high MSS score correlated with a higher immune infiltrate in the tumors and related to the higher expression of MSS in tumor-infiltrating microglia. Importantly, the MSS score could be predictive of MAPKi therapy response and could be a future tool to stratify patients.

We thank the Reviewer 4 for their precise review of our manuscript and their general positive appreciation of our work.

Comment 13:

Some remarks remain:

- An aspect that needs clarification is the fact that the four MSSs were established in vitro and based on cell line studies. However, ultimately, and somewhat surprisingly in the flow of the story, it turns out that tumor-infiltrating microglia may be important mediators of the MAPKi responsiveness in CNS tumors. Since microglia were not present in the initial screening for signatures, it seems to me that important signature genes may actually be missing and that these scores could be improved if signatures would have been established based on in vivo (eg PDX) data.

We thank Reviewer 4 for raising this very interesting and challenging point.

While we fully agree that, *a posteriori*, it would make sense to generate the scores from *in vivo* samples, it was less evident to have done so *a priori*. Theoretically, one would expect that only the tumor population is sensitive to MAPKi, since only the tumor cells harbor the genetic MAPK alteration. Hence the rationale to start from pure tumor cell populations from *in vitro* testing datasets. While we did not expect the microglia population to appear as potential mediator of MAPKi effects *a priori*, the actual extent of the microglia population's role in MAPKi response remains to be explored, as discussed in the response to the following comment (Comment 14) and also mentioned in our discussion on line 557-591.

By design, our signatures were derived focusing on MAPK-related genes only (Hallmark_Kras_UP signature), in order to avoid the inclusion of any genes that would be specific to the tissue the tumor cell lines were derived from. By doing so, we aimed at generating MAPKi sensitivity signatures that would be free of any tissue/cell type-specific genes (as confirmed by our confoundR analysis), to make the signatures at least potentially applicable to any cell type.

In addition, establishing signatures from PDX or pLGG patient data including both tumor and microglia cells represents a significant challenge, as 1) there is no PDX model recapitulating the true pLGG biology,¹⁰ and 2) there is currently no large dataset derived from patient data with known response to MAPKi therapy available (see our response to comment 1 from reviewer 1).

As an alternative approach, we have investigated the PDX dataset from the XenaDB, with the limitations that 1) it does not contain brain tumors (i.e. no PDX with brain tissue microglia) and in particular no pLGGs, and 2) PDX were transplanted in immune-compromised NSG mice. However, such PDX might have retained the tumor-associated macrophages which, as discussed by reviewer 4 in their last comment (Comment 18), in non-CNS tumors might play a role similar to microglia.

We generated PDX-derived MSS for the BRAFi Type I½ encorafenib, and the MEKi trametinib and binimetinib using the XevaDB PDX-dataset. To make sure that the PDX samples contained some immune cells, we first evaluated the estimated proportion of immune cells using the ESTIMATE-derived immune score. We show (Fig. 7) that the primary material used to transplant mice (i.e. before treatment) in the MEKi cohorts (binimetinib, trametinib) retain a significant proportion of immune cells compared to the samples used in the BRAFi Type I½ cohort (encorafenib).

Fig. 7 Immune score measure in primary samples (before treatment) from all 3 treatment cohorts

We then generated PDX-derived gene signatures for encorafenib, binimetinib and trametinib. There was only a small overlap between the genes comprised in the PDX-derived MSS for the BRAFi encorafenib and the MEKis trametinib and binimetinib, and those comprised in our original MEKi and BRAFi MSS (Fig. 8).

Fig. 8 Gene signatures overlap

We then applied these new signatures in our patient datasets to evaluate the predictive power of the PDX-derived MSS compared to our MSS (Fig. 9). In the pLGG samples treated with the MEKi trametinib, the PDX-derived signature for the MEKi (Fig. 9A and B, respectively) did not outperform our cell line-derived MSS (Fig. 9C).

Fig. 9 Analysis of PDX-derived MSSs for the MEKi trametinib and binimetinib in pLGG samples from patients treated with the MEKi trametinib

Importantly, when comparing the prediction made by the PDX-derived binimetinib signature (Fig. 10A) to the predicted proportion of immune infiltration (ESTIMATE score, Fig. 10B), we can see a clear association of these two signatures, suggesting a potential confounding of the PDX-derived score by immune-related genes, resulting in a better prediction of the proportion of immune infiltration than the actual response to the MEKi.

Fig. 10 Analysis of PDX-derived MSS for the MEKi binimetinib and the ESTIMATE immune infiltration in pLGG samples from patients treated with the MEKi trametinib

In addition, in a second independent cohort of melanoma samples with mutually exclusive BRAF fusion alterations and treated with the BRAFi Type I½ vemurafenib (used to answer Reviewer 1's comment), the PDX-derived BRAFi Type I½ signature (Fig. 11A) also did not outperform our cell line-derived MSS (Fig. 11B).

Fig. 11 Analysis of PDX-derived MSSs for the BRAFi Type I 1/2 the the BRAFi Type I 1/2 MSS in melanoma samples from patients treated with the BRAFi Type I 1/2 vemurafenib

Taken together, the data suggest that 1) there is unfortunately not perfect dataset to derive gene-based sensitivity signatures from pLGG in vivo data, or pLGG patients treated with MAPKi paired with RNA sequencing data at baseline, 2) our MSS signatures contain enough genes to be able to predict MAPKi sensitivity in patient samples and is not outperformed by the PDX-derived signatures, and 3) adding genes specific to a certain cell population (such as tumor-infiltrating immune cells) has the risk of potentially introducing a confounding effect. For these reasons, we favor to not further add genes to our signatures. This is now briefly discussed in the manuscript line 434-435 “Of note, PDX-derived sensitivity signature analysis did not yield signatures superior to our MSS (see suppl. Mat. And Suppl. Figure S24).” And more extensively discussed in the supplementary materials: “Since our MSSs were derived from cell lines, i.e. without microenvironmental cells, we verified whether key genes related to sensitivity in the microenvironment cells might be missing and could be included in order to improve the MSSs prediction power. To alleviate the lack of data from PDX model recapitulating the true pLGG biology,⁷² and lack of large dataset derived from patient data with known response to MAPKi therapy, we generated PDX-derived MSS for the BRAFi Type I 1/2 encorafenib, and the MEKi trametinib and binimetinib using the XevaDB PDX-dataset. We first show that the PDX samples used in the MEKi cohorts (binimetinib, trametinib) retains a significant proportion of immune cells (probably from the primary material the PDX is derived from), while the PDX samples used in the BRAFi Type I 1/2 cohort (encorafenib) retained a significantly smaller proportion of immune cells (Suppl. Fig. S24A). We could generate PDX-derived sensitivity gene signatures for encorafenib (Enco-PDX MSS), binimetinib (Bini-PDX MSS) and trametinib (Tram-PDX MSS) by selecting the genes contributing to the enrichment edge of the “Hallmark_KRAS_SIGNALING_UP” gene signature in the responsive samples (tumor volume reduction upon treatment) compared to the non-responsive samples (tumor volume increase upon treatment). There was only a small overlap between the genes comprised in the PDX-derived MSS and those comprised in our original MSS (Suppl. Fig. S24B, Suppl. Table S4). We then applied these new signatures in our pLGG patient datasets. In the pLGG samples treated with the MEKi trametinib, the Tram-PDX MSS correlated with treatment response in a similar than our MEK1/2i MSS (Suppl. Fig. S24C and Fig. 4D, respectively). The Bini-PDX MSS poorly predicted treatment response (Suppl. Fig. S19D), and had a similar pattern than the predicted immune infiltration (Suppl. Fig S22). In the cohort of melanoma samples with mutually exclusive BRAF fusion alterations and treated with the BRAFi Type I 1/2 vemurafenib, the Enco-PDX MSS did not outperform our cell line-derived MSS (Suppl. Fig. S24E). Taken together, the data suggest that our MSS signatures

contains enough genes to be able to predict MAPKi sensitivity with high enough efficacy in patient samples, and that adding genes specific to a certain cell population has the risk of potentially introducing a confounding effect.”, and the aforementioned data have been included in supplementary figure S24. We also discuss this data in the discussion line 511-516 “In particular, our signatures were derived focusing on MAPK-related genes only (Hallmark_Kras_UP signature), in order to avoid the inclusion of any genes that would be specific to the tissue the tumor cells were derived from. By doing so, we aimed at generating MAPKi sensitivity signatures that would be free of any tissue/cell type-specific genes (as confirmed by our confoundR analysis and PDX-derived MSS analysis), to make the signatures virtually applicable to any cell type.”

Comment 14:

- Related to this, it would be nice to establish the relative importance of MSSs within microglia versus within the cancer cells in relation to MAPKi responsiveness. Can the authors find patients in which MSS is high in cancer cells, but not in microglia/macrophages, or vice versa? Would this lead to a different outcome for the MAPKi treatment? The essential question here is obviously: which cells are actually the crucial ones to mediate MAPKi responses?

This is another very stimulating point that Reviewer 4 is making here. We fully agree that identifying which cell population(s) are crucial to mediate MAPKi response (tumor cells, microglia, both?) is of high relevance here.

First of all, we could not find patients in which the MSS was higher in the tumor cells compared to microglia/macrophages, or vice versa. As shown in Figure 6B and 8C, the tumor and microglia compartments have very similar MSS overall. From Fig. 8C, the samples PA3 and PA4 might look like the scores are higher in the microglia/macrophages compartment, however this is probably because of the small proportion of tumor cells in these samples. But our cohort is small, so it cannot be excluded that such a scenario might occur in a larger cohort.

Nonetheless, the implications this could have on treatment outcome are very interesting. To investigate this, we performed a ssGSEA on our primary pLGG samples using immune signature from the ESTIMATE, and correlated the results with the observed response to MEKi therapy (as shown in our response to the previous comment 13) (Fig. 10B, shown again below).

An increased signature score for microglia does not correlate with best MAPKi response type in this small data set. Overall, the samples with a higher predicted immune infiltration were those with a better outcome, but the relationship was not linear. This may suggest that the immune infiltration plays a role in the treatment response, but does not overrule the role played by the tumor compartment in MAPKi response, as suggested by Fig. 6B, where the MSS is found high in both cell populations. We hypothesize that the combination of both cell populations and their interaction is crucial to mediate response to MAPKi in pLGG.

This is now discussed in the manuscript line 394-402 “Interestingly, the predicted immune infiltration (ESTIMATE immune score) did not correlate with best MAPKi response type (Suppl. Fig. S22). Overall, the samples with a higher predicted immune infiltration were those with a better outcome, but the relationship was not linear. This suggest that the immune infiltration may play a role in the treatment response, but does not overrule the contribution played by the tumor compartment in MAPKi response, as suggested by Fig. 6B, where the MSS is found high in both cell populations.

These data suggest that the microenvironment, and particularly tumor-associated microglia, could be involved, along with the tumor compartment, in the degree of sensitivity towards MAPKi, and might represent a target of MAPKi in pLGG.” , and the data was included in supplementary figure S22.

Comment 15:

- Fig 6A. CX3CR1 is not bad, but TMEM119 is a somewhat "cleaner" marker to identify microglia.

We thank Reviewer 4 for giving his expertise on the markers used to identify our cell populations in a more accurate way. We had previously described TMEM119 expression in the supplementary materials (see supplementary figure S16). Motivated by the reviewer's comment, we have now swapped marker plots, and have now included the TMEM119 UMAP to the main figure.

Comment 16:

No cDC1 were detected (XCR1 expression)?

We were also able to detect cDC1 positive cells, the data is shown in supplementary figure S16.

Comment 17:

Did you consider cells that co-express CD3G and PRF1, as these are likely NKT cells?

We could also detect CD3G and PRF1-expressing NKT cell populations, but due to largely overlapping marker gene expression profiles, they could not be separated from neighboring NK and CD8+ cytotoxic T-cell populations (Fig. 12B).

Fig. 12 In-depth cell population characterization of the scRNAseq data from 6 primary pLGG samples

Since our data shows no indications that NKT cells show a particular sensitivity to MAPKi in pLGG, we refrained from a more detailed investigation and did not include this data. However, in case the reviewer and/or editors would find this data necessary to be added, we would be happy to include them in the final version of the manuscript.

Comment 18:

- Statements in lines 381-385 require some nuance. MSSs are clearly also high in monocyte-derived TAMs, albeit somewhat less high than in tumor-associated microglia. Hence, MSS is not "specifically" elevated in tumor-associated microglia, and it can not be excluded that monocyte-derived TAMs may contribute to MAPKi efficiency, especially in non-CNS tumors. It is also true that the correlation between MSSs and ESTIMATE is highest in CNS tumors, but there is never a significant difference with non-CNS tumors. In conclusion, statements need to be tempered and other possibilities need to be acknowledged.

Reviewer 4 makes a very good point with this comment, and we gladly nuanced our statement following the reviewer's advice. It is indeed conceivable that tumor-associated macrophages might play an equivalent role to the tumor-associated microglia in non-CNS

tumors, which brings very interesting perspectives for future studies in such tumor entities. We have now adjusted the statement in line 441-445 “Of note, this does not rule out a possible role played by tumor-associated macrophages (TAMs), which also present high MSS (albeit lower compared to microglia) in our pLGG scRNAseq dataset (Fig. 4B). Since the difference between CNS and non-CNS tumor doesn’t reach significance, it can not be excluded that monocyte-derived TAMs may contribute to MAPKi response, especially in non-CNS tumors.”.

References

1. Sigaud, R. *et al.* The first-in-class ERK inhibitor ulixertinib shows promising activity in mitogen-activated protein kinase (MAPK)-driven pediatric low-grade glioma models. *Neuro. Oncol.* (2022) doi:10.1093/NEUONC/NOAC183.
2. Bongaarts, A. *et al.* The coding and non-coding transcriptional landscape of subependymal giant cell astrocytomas. *Brain* **143**, 131–149 (2020).
3. Dry, J. R. *et al.* Transcriptional Pathway Signatures Predict MEK Addiction and Response to Selumetinib (AZD6244). *Cancer Res.* **70**, 2264–2273 (2010).
4. Parca, L. *et al.* Modeling cancer drug response through drug-specific informative genes. *Sci. Rep.* **9**, (2019).
5. Li, Y. *et al.* Predicting tumor response to drugs based on gene-expression biomarkers of sensitivity learned from cancer cell lines. *BMC Genomics* **2021** **22**, 1–18 (2021).
6. Burotto, M., Chiou, V. L., Lee, J. M. & Kohn, E. C. The MAPK pathway across different malignancies: A new perspective. *Cancer* **120**, 3446 (2014).
7. Carter, C. A. *et al.* Selumetinib with and without erlotinib in KRAS mutant and KRAS wild-type advanced nonsmall-cell lung cancer. *Ann. Oncol.* **27**, 693–699 (2016).
8. Gandara, D. R. *et al.* A Phase 1/1b Study Evaluating Trametinib Plus Docetaxel or Pemetrexed in Patients With Advanced Non–Small Cell Lung Cancer. *J. Thorac. Oncol.* **12**, 556–566 (2017).
9. Van Laethem, J. L. *et al.* Phase I/II Study of Refametinib (BAY 86-9766) in Combination with Gemcitabine in Advanced Pancreatic cancer. *Target. Oncol.* **12**, 97–109 (2017).
10. Chiacchiarini, M. *et al.* Pediatric low-grade gliomas: molecular characterization of patient-derived cellular models. *Child's Nerv. Syst.* **37**, 771–778 (2021).

REVIEWERS' COMMENTS

Reviewer #1 (Remarks to the Author):

The authors have addressed all my concerns.

I would stress more in the text (and maybe in the title) the possible applications also for human melanoma patients since the clinical relevance of MSS has been tested in melanoma patients (Fig S14).

Reviewer #2 (Remarks to the Author):

The authors have extended their study performing selectively tests on three additional drugs. As a suggestion for future work I believe a more comprehensive analysis of the whole GDSC dataset which includes hundreds of compounds would lead to deeper discoveries and better validation of the approach.

Reviewer #3 (Remarks to the Author):

The authors have appropriately responded to the comments and I'm happy with the final manuscript.

Reviewer #4 (Remarks to the Author):

The authors sufficiently addressed my concerns and adapted their manuscript accordingly

Response to reviewers' comments on NCOMMS-22-52491A

Reviewer #1, expertise in paediatric brain cancer genomics

The authors have addressed all my concerns.

I would stress more in the text (and maybe in the title) the possible applications also for human melanoma patients since the clinical relevance of MSS has been tested in melanoma patients (Fig S14).

We thank Reviewer #1 for their positive feedback and confirmation that we have addressed all their concerns. We furthermore appreciate Reviewer #1's final comment. We have now added a sentence in the discussion to emphasize the possible application for melanoma patients, in line 533-535: "This indicates that, beyond pLGG, the MSSs described here could potentially also be applied to predict MAPKi sensitivity in melanoma patients with mutually exclusive MAPK alterations".

Reviewer #2, expertise in scRNA-seq, bioinformatics and drug response

The authors have extended their study performing selectively tests on three additional drugs. As a suggestion for future work I believe a more comprehensive analysis of the whole GDSC dataset which includes hundreds of compounds would lead to deeper discoveries and better validation of the approach.

We would like to thank Reviewer #2 for their valuable feedback on our study. We acknowledge the suggestion for future work, emphasizing the importance of a more comprehensive analysis of the entire GDSC dataset. We agree that such an expanded investigation may have the potential to provide further validation of our approach.

Reviewer #3, expertise in MAPK signalling, kinase inhibitors and GBM therapy

The authors have appropriately responded to the comments and I'm happy with the final manuscript.

We appreciate the positive feedback from Reviewer #3 and are delighted to hear that they are satisfied with the final manuscript.

Reviewer #4, expertise in TiME in GBM

The authors sufficiently addressed my concerns and adapted their manuscript accordingly.

We thank Reviewer #4 for their feedback and for acknowledging that we have sufficiently addressed their concerns.

Additional note:

We were able to correct a mistake in the graphical depiction in Supplementary Figure S2. This doesn't change the interpretation nor the results of our study.